# Gas chromatography vs. quantum cascade laser-based N₂O flux measurements using a novel chamber design

Christian Brümmer[1,*], Bjarne Lyshede[1], Dirk Lempio[1], Jean-Pierre Delorme[1], Jeremy J. Rüffer[1,*], Roland Fuß[1], Antje M. Moffat[1], Miriam Hurkuck[1], Andreas Ibrom[2], Per Ambus[3], Heinz Flessa[1], Werner L. Kutsch[1,4]

[1]Thünen Institute of Climate-Smart Agriculture, Braunschweig, Germany
[2]Department of Environmental Engineering, Technical University of Denmark, Lyngby, Denmark
[3]Department of Geosciences and Natural Resource Management, University of Copenhagen, Denmark
[4]Integrated Carbon Observation System, ICOS ERIC Head office, Helsinki, Finland
[*]previously published under the name: Jeremy Smith

*Correspondence to*: Christian Brümmer (christian.bruemmer@thuenen.de)

**Abstract.** Recent advances in laser spectrometry offer new opportunities to investigate soil-atmosphere exchange of nitrous oxide. During two field campaigns conducted at a grassland site and a willow field, we tested the performance of a quantum cascade laser (QCL) connected to a newly developed automated chamber system against a conventional gas chromatography (GC) approach using the same chambers plus an automated gas sampling unit with septum capped vials and subsequent laboratory GC analysis. Through its high precision and time resolution, data of the QCL system were used for quantifying the commonly observed non-linearity in concentration changes during chamber deployment, making the calculation of exchange fluxes more accurate by the application of exponential models. As expected, the curvature in the concentration increase was higher during long (60 min) chamber closure times and under high flux conditions ($F_{N2O}$>150 µg N m⁻² h⁻¹) than those that were found when chambers were closed for only 10 min and/or when fluxes were in a typical range of 2 to 50 µg N m⁻² h⁻¹. Extremely low standard errors of fluxes, i.e. from ~0.2 to 1.7 % of the flux value, were observed regardless of linear or exponential flux calculation when using QCL data. Thus, we recommend reducing chamber closure times to a maximum of 10 min when a fast-response analyzer is available and this type of chamber system is used to keep soil disturbance low and conditions around the chamber plot as natural as possible. Further, applying linear regression to a 3-min data window with rejecting the first two minutes after closure and a sampling time of every 5 s proved to be sufficient for robust flux determination assuring standard errors of N₂O fluxes still being on a relatively low level. Despite low signal to noise ratios, GC was still found to be a useful method to determine mean soil-atmosphere exchange of N₂O at longer time scales during specific campaigns. Intriguingly, the consistency between GC and QCL-based campaign averages was better under low than under high N₂O efflux conditions, although single flux values were highly scattered during the low efflux campaign. Furthermore, the QCL technology provides a useful tool to accurately investigate the highly debated topic of diurnal courses of N₂O fluxes and its controlling factors. Our new chamber design prevents the measurement spot from

unintended shading and minimizes disturbance of throughfall, thereby complying with high quality requirements of long-term observation studies and research infrastructures.

## 1 Introduction

Accurate determination of ambient nitrous oxide ($N_2O$) concentrations and the associated exchange between soil and atmosphere has been in the focus of environmental research for several years. Nitrous oxide is of high relevance for the Earth's greenhouse gas budget due to its long residence time in the troposphere and its relatively large energy absorption capacity per molecule, resulting in a cumulative radiative forcing almost 300 times higher than the same mass unit of carbon dioxide over a 100-year period when climate-carbon feedbacks are included (IPCC, 2013). It is predominantly emitted as a byproduct of nitrification and an intermediate product of denitrification and nitrifier denitrification, which are key microbiological processes in the soil nitrogen (N) cycle (Firestone and Davidson, 1989; Wrage et al., 2001; Thomson et al., 2012; Butterbach-Bahl et al., 2013). Main $N_2O$ sources are agricultural activities in the form of N fertilization. In smaller quantities, $N_2O$ is also produced through biomass burning, degassing of irrigation water, and industrial processes (Seinfeld and Pandis, 2006). On the other hand, some field studies report that soils can also consume $N_2O$, albeit the strength of this sink has not yet been thoroughly evaluated (Donoso et al., 1993; IPCC 2007; Chapuis-Lardy et al., 2007).

Precise measurements of $N_2O$ – particularly at the field scale – are therefore essential for specific applications in ecosystem research such as the study of N cycling, fertilization effects, and for the compilation of full greenhouse gas budgets. The most common method to measure soil-atmosphere exchange of $N_2O$ is the operation of static chambers (Hutchinson and Mosier, 1981; Schiller and Hastie, 1996). The $N_2O$ flux is calculated from the concentration increase (or decrease) over time in a gas-tight chamber, which is usually attached to a collar that is permanently inserted into the soil. A number of approaches have emerged over the last years where the air sample is either manually collected using a syringe through a septum and/or directly inserted into sample vials (e.g., Castaldi et al., 2010; Jassal et al., 2008, 2011; Livesley et al., 2011; Lohila et al., 2010; Parkin and Venterea, 2010 and references therein) with subsequent analysis on gas chromatography (GC) systems using [63]Ni electron capture detectors for $N_2O$ detection. Different chamber designs and air sampling procedures exist, either with manual, semi-automated, i.e. automatic sampling but manual transport of air samples in syringes or vials to the GC (this study), or fully automated gas collection, where the air samples are directly pumped (or sucked) via carrier gas to a temperature-stable housing equipped with a GC in the field (e.g., Brümmer et al., 2008; Butterbach-Bahl et al., 1997; Dannenmann et al., 2006; Flessa et al., 2002; Papen and Butterbach-Bahl, 1999; Rosenkranz et al., 2006).

In the last decade, substantial progress has been made in the development of fast-response technologies for analyzing a variety of N and carbon (C) trace gases. These are tunable diode laser absorption spectrometers (TDLAS), quantum cascade lasers (QCL) and devices originating from individual applications such as Fourier transform infrared (FTIR) spectrometers or custom-made converters coupled to chemiluminescence detectors (CLD). These robust, fast and precise analyzers are essential for long-term monitoring of biosphere-atmosphere exchange and have even allowed first eddy covariance (EC)

measurements of field-scale $N_2O$, methane ($CH_4$) (e.g., Rinne et al., 2005; Denmead et al., 2010; Kroon et al., 2010; Neftel et al., 2010; Tuzson et al., 2010; Jones et al., 2011; Merbold et al., 2014), and reactive N fluxes (Horii et al., 2004; Ammann et al., 2012; Brümmer et al., 2013). Continuous observations of trace gas exchange over time scales from hours to decades enable researchers to evaluate diurnal, seasonal and interannual variability and trends as well as the elucidation of climatic

and management controls on gas exchange patterns (e.g., Baldocchi et al., 2001; Brümmer et al., 2012; Kutsch et al., 2010). With regard to chamber measurements, it is expected that the precision and time resolution of the above-mentioned technologies may considerably reduce the chamber closure duration for single flux measurement events, thereby minimizing plot disturbance and allowing for a significant increase in repeated measurements leading to more robust databases, which are required for reliable greenhouse gas budgets. Although the EC methodology provides near-continuous time series of

greenhouse gas concentrations and exchange, chamber measurements will certainly still be required in the future as prerequisites for EC measurements are sometimes not fulfilled (for example through insufficient turbulent mixing, complex terrain, inhomogeneous fetch) and small-scale spatial variability or emissions from replicated field plot experiments can only be determined by chamber measurements. Some first examples of high-resolution chamber measurements using fast-response analyzers can be found in Cowan et al. (2014a; 2014b), Hensen et al. (2006), Laville et al. (2011), Sakabe et al.

(2015), and Savage et al. (2014).

The comparability, applicability and uncertainty associated with the respective approach are currently debated in the ecosystem research community, e.g. when comparing fluxes from GC/vial systems with those from more recent continuous setups such as QCL systems. In this context, the flux determination method was found to be an important factor (e.g., Kroon et al., 2008; Forbrich et al., 2010). Fluxes are often calculated using a linear regression of the change in headspace

concentration over time and are scaled to the collar area, including a temperature and pressure correction (e.g. Savage et al., 2014). However, several other studies demonstrate the need for non-linear models for soil-atmosphere trace gas flux estimation (Hutchinson and Mosier, 1981; Livingston et al., 2006; Kutzbach et al., 2007; Kroon et al., 2008; Pedersen et al., 2010; Pihlatie et al., 2013). It has been argued that molecular diffusion theory states that chamber effects lead to declining gradients in the relationship between concentration and time and that slight chamber leakages create the same effect

(Hutchinson and Mosier, 1981; Livingston et al., 2006; Pedersen et al., 2010). Nevertheless, linear concentration data often predominates (e.g. Forbrich et al., 2010), which may not necessarily be in conflict with the theory as non-linearity is sometimes not visible in data series with only a limited number of samples (mostly due to noisy concentration measurements or effects of small chambers; Pedersen et al., 2010).

To further investigate effects of flux estimation methods on the one hand and the use of different gas analyzer types on the

other hand, our study comprises $N_2O$ chamber flux measurements from two campaigns conducted with a newly developed chamber system under different environmental conditions. The aims of this study were as follows:

- Presentation of a novel chamber design that is connected to both a vial air-sampling setup with subsequent GC analysis and a QCL spectrometer.

- Characterization of the shape of the concentration increase/decrease to identify whether $\partial c/\partial t$ is rather linear or non-linear including a quantification of the curvature ($\kappa$) in concentration increase/decrease (Section 3.1). The parameter $\kappa$ was further used to verify chamber sealing by checking its dependency on wind speed, wind direction, on the flux itself and on closure time.
- Comparison of $N_2O$ fluxes and their associated standard errors from linear and non-linear regression models (Section 3.2).
- Testing the novel chamber system under high and low flux conditions and comparing GC vs. QCL-based flux estimates (Section 3.3).
- Investigation of ecosystem and climate-specific flux characteristics such as $N_2O$ uptake and diurnal variation (Section 3.4).

## 2 Methods

### 2.1 Chamber design

Nitrous oxide measurements were carried out using a newly developed semi-automatic chamber system (Figure 1). It consisted of aluminum guiding racks (length 2121 mm, width 936 mm, height 3033 mm) with aluminum soil collars (length 750 mm, width 750 mm, height 160 mm, inserted 0.10 m into the soil), and opaque PVC chambers (color white, interior dimensions: length 777 mm, width 777 mm, height 565 mm) (Ps-plastic, Eching, Germany). Subtracting inside items such as an axial fan, screws, supporting racks and tubes, the chambers have a headspace volume of 0.33 $m^3$ and covered a surface area of 0.56 $m^2$. Depending on vegetation height, extension modules (interior dimensions: length 730 mm, width 730 mm, height 360 mm) can be connected to the chambers (total headspace volume was then 0.54 $m^3$) if needed over taller vegetation, but were not used in this study. EPDM-gaskets (20 mm x 15 mm) were attached to the bottom of each chamber in an aluminum u-channel to ensure gas tight closure when chambers were operating. Up to three chambers can be combined to one system (Figure 1B) with a joint control unit and autosampler or analyzer. Two custom made temperature probes (Pt100) were installed inside and outside of each chamber to measure ambient air temperatures. Chambers were ventilated during measurements using an axial fan, which was mounted to produce a horizontally oriented airflow alongside chamber walls to minimize interference with the natural steady-state soil efflux, but to maximize proper mixing of the chamber headspace as was described in Drösler (2005). The air was sampled from the top centre of the lids. Chamber operation was controlled by a logic module (Millenium 3, Crouzet, Hilden, Germany). An autosampler consisting of a membrane pump (operated at 0.8 L min$^{-1}$, NMP 830 KNDC, KNF Neuberger, Freiburg, Germany), an absorber to avoid water condensation within tubes (3.2 mm ID, 6.4 mm OD) (BEV-A-Line, ProLiquid GmbH, Überlingen, Germany) and valves as well as an exchangeable rack for 162 headspace vials (20 mL, WICOM WIC 43200, Maienfeld, Germany) were connected to the chamber system. Chambers were lifted and moved down by a 24 V (DC) motor winch and were directed to the soil collar by

the aluminum rack. After measurement events, the chamber was lifted to 1.18 m above ground and dragged backwards in a 45° angle to keep the soil and vegetation inside the soil collar under as natural conditions as possible (e.g. prevention of shading and undisturbed throughfall). To avoid pressure changes when setting the chamber on the collar, the chamber had a 1.5 m pressure compensation tube leading from the inside through the side wall of the chamber to the outside. Information about our chamber system including the construction plan is open to the scientific community and can be requested from the authors.

## 2.2 Campaigns and measurement setup

Two field campaigns were conducted in fall 2012 in Braunschweig, Germany, and in spring 2013, at Risø Campus, Technical University of Denmark, using both GC and QCL chamber setups (see Table 1 for additional information). The chamber architecture was identical during the two campaigns. Sites and time periods were selected with the aim to compare chamber system performance under high and low flux conditions. Due to low temperatures and no fertilizer applied, we expected a low exchange regime during the Braunschweig campaign, whereas higher fluxes were expected at Risø (higher temperatures and a substantial amount of fertilizer applied).

During parallel operation of GC and QCL, chambers were closed for 60 minutes at both sites to measure the concentration increase. When only QCL measurements were conducted, i.e. at Risø at DOY <105.5 and DOY >108.5, chambers were closed for only 10 minutes. For the GC setup, four air samples were taken at 0, 20, 40, and 60 minutes after chamber closure to calculate one flux rate. Air samples (20 mL) were pumped through the tubing system using a membrane pump (3.2 L min$^{-1}$, NMP 830 KNDC, KNF Neuberger, Freiburg, Germany) and were injected into septum-capped vials. Two cannulas were automatically inserted through the septum, one cannula acting as sample air inlet until overpressure was established, and the other cannula acting as outlet for cycling the air back to the chamber. Air samples were stored in the exchangeable rack of the autosampler unit and were analyzed in the GC-lab of the Thünen Institute using a GC-2014 (Shimadzu, Duisburg, Germany; modified according to Loftfield et al., 1997) with an electron capture detector for $N_2O$ analysis. Performance of the GC-system was checked weekly by conducting ten consecutive measurements of a standard gas with ambient $N_2O$ concentration (320 ppb). Samples were only analyzed if the coefficient of variation of peak areas during this test was below 3 %.

Parallel to the autosampler setup for GC analysis, we operated our chamber system directly connected to a QCL (continuous-wave quantum cascade laser absorption spectrometer, model mini-QCLAS, Aerodyne Research Inc., Billerica, Massachusetts, USA; see Nelson et al. (2004) for principle of operation) in a thermo-controlled housing. Briefly, the laser is thermoelectrically cooled (Thermocube) to 25 °C, uses a 76-m path length, 0.5 L volume and multiple pass absorption cell for sampling, and operates at 40 Torr. It provides a measurement precision of 0.04 ppb (1$\sigma$) within an averaging interval of one second. Calibration is performed by continuously aligning the $N_2O$ absorption peak of the sampled air to the standard of the HITRAN database (Rothman et al., 2009). A dry vacuum scroll pump (BOC Edwards XDS10, Sussex, UK) maintained a steady flow rate of 1.0 L min$^{-1}$. After passing the QCL cell, the sample air was cycled back to the respective chamber to

avoid underpressure conditions and unintentional sucking of soil air into chambers. Data was stored on the QCL's internal hard drive at a frequency of 10 Hz.

The detection limit (LoD) of our QCL and GC setups could be estimated using our campaign data assuming stationary conditions during the low flux campaign in Braunschweig. Taking the whole campaign into account, the calculated standard deviations were 2.5 µg m$^{-2}$ h$^{-1}$ and 7.5 µg m$^{-2}$ h$^{-1}$ for QCL and GC measurements, respectively. Thus, the resulting 2-$\sigma$ uncertainty range for QCL was 5.0 µg m$^{-2}$ h$^{-1}$ and for GC 15.0 µg m$^{-2}$ h$^{-1}$. If only the first quarter of the Braunschweig campaign data are taken, i.e., a period where environmental conditions were less variable than over the whole campaign, the calculated standard deviations were 1.3 µg m$^{-2}$ h$^{-1}$ and 6.5 µg m$^{-2}$ h$^{-1}$ for QCL and GC measurements, respectively. Thus, the resulting 2-$\sigma$ uncertainty range for QCL was 2.6 µg m$^{-2}$ h$^{-1}$ and for GC 13.0 µg m$^{-2}$ h$^{-1}$. These estimates can be regarded as an upper flux detection limit. A supposable lower flux detection limit solely depends on the sensitivity of the analyzers. Precision of the QCL is 0.03 and 0.01 ppb when averaging over 1 and 60 s, respectively. Table 2 summarizes features of the chamber-analyzer system used in this study.

## 2.3 Flux calculation

GC-based N$_2$O fluxes using linear, robust linear (Huber, 1981), and modified Hutchinson-Mosier regression (HMR; *cf*. Pedersen et al., 2010) were calculated as described in Leiber-Sauheitl et al. (2014) after converting molar concentrations into mass concentrations using temperature but no pressure correction. Briefly, non-linear flux estimation with the HMR method (R Core Team, 2012; HMR package version 0.3.1) was performed when four data points were available and all of the following criteria were met, i.e. (1) the HMR function could be fitted, (2) Akaike information criterion (AIC; Burnham and Anderson, 2004), which is a measure of (relative) model quality, i.e., gives fit quality penalized by the model's degrees of freedom, and can be used to compare the quality of different model fits to the same dataset, was lower for HMR fit than for linear fit, (3) p value of flux calculated using HMR was lower than that from robust linear fit, and (4) the HMR flux was less than four times larger than the robust linear flux. Otherwise, robust linear regression or ordinary linear regression was used when four or three data points were available, respectively.

QCL-based fluxes were estimated using two different methods. We applied the non-linear HMR model with a slightly modified parameterization (Equation 1 this study; *cf*. Moffat, 2012) to the 60-min dataset of a full chamber cycle (10-min cycle in Risø at DOY <105.5 and DOY >108.5) and compared these fluxes with those resulting from an application of linear regression when only the first three minutes of data after chamber closure were used (*cf*. Section 3.2).

To investigate the frequently observed non-linearity in chamber field data, we computed a quantitative parameter $\kappa$ describing the curvature in N$_2$O concentration increase (or decrease) over time (60-min and 10-min QCL data only). Based on the assumption of exponential gas concentration changes in the chamber (*cf*. Nakano et al., 2004) using

$$c(t) = c_{max}\left(1 - \exp\left(\frac{-k}{c_{max}}t\right)\right) + c_0 \qquad (1)$$

with $c(t)$ being the $N_2O$ concentration in the chamber at a certain point in time, $c_{max}$ the maximum possible concentration, $c_0$ the measured concentration at $t=0$, and $k$ the initial flux $F_0$ divided by the effective chamber height $h$, we estimated the $N_2O$ soil-atmosphere flux as the first derivative of Equation (1) evaluated at $t=0$, i.e.

$$c'(t)|_{t=0} = k \tag{2}$$

and the curvature parameter $\kappa$ as the second derivative of Equation (1) evaluated at $t=0$, i.e.

$$c''(t)|_{t=0} = -\frac{k^2}{c_{max}} = \kappa. \tag{3}$$

Units for concentrations $c(t)$, $c_{max}$, and $c_0$ are g m$^{-3}$, units for $k$ are g m$^{-2}$ s$^{-1}$, and units for $\kappa$ are g m$^{-3}$ s$^{-2}$. Negative values of $\kappa$ correspond to concave curvature indicating a plateauing, i.e. saturating concentration increase over time. Standard errors in this study were calculated as the parameter errors from the respective regression model with the algorithm being based on the Levenberg-Marquardt method ('nlsLM function in R package 'minpack.lm', R Core Team, 2012). Standard errors are solely associated with the flux calculation method and not with any kind of observational errors or issues related to measurement performance such as changes in flow rate, temperature sensitivity of the QCL, pump performance, or changes in chamber volume due to rough soil surfaces or plants in the chamber.

## 3 Results

### 3.1 Shape of concentration increase and curvature ($\kappa$) determination

Significantly different patterns in chamber $N_2O$ concentration changes during the Braunschweig and Risø campaigns were observed (Figure 2). While increases in the order of 10 to 20 ppb per hour (one chamber cycle) were found for the grassland site in Braunschweig, steep concentration increases measured on the harvested willow field at Risø were almost exclusively higher than 100 ppb per hour and reached maximum rates of over 650 ppb per hour in the period from DOY 105.5 to DOY 108.5. For the low exchange regime in Braunschweig, GC-based data points were highly scattered and rarely showed a clear increasing (or decreasing) tendency making flux calculations difficult. For the high exchange regime at Risø, GC-based concentration data were mostly showing well-defined increases and were similar to those obtained by the QCL system (*cf.* Section 3.3). The latter showed a precise and robust performance with clear base line levels and obvious chamber cycles during both campaigns. None of the QCL-based measurements revealed concentration decreases, i.e. negative fluxes ($N_2O$ uptake) while chambers were closed.

Results of the investigation on quantifying the curvature in $c(t)$, expressed as $\kappa$, are given in Figure 3. Extremely low absolute $\kappa$ values between $-10^{-4}$ and $-10^{0}$ – indicating quasi linearity in $\partial c/\partial t$ – were almost exclusively found under low flux conditions, whereas fluxes $>100$ µg N m$^{-2}$ h$^{-1}$ were only observed when $\kappa$ was $< -10^{1}$ (Figure 3A).

## 3.2 Comparison of N$_2$O fluxes and their associated errors from linear and non-linear regression models

With the QCL's high time resolution – in this study operated at the analyzer's maximum frequency of 10 Hz – we compared N$_2$O flux estimates based on 60-min (DOY 105.5 to DOY 108.5) and 10-min (DOY <105.5 and DOY >108.5) closure periods calculated by the modified HMR approach with those flux estimates that are based on the first three minutes of concentration data only and were calculated by linear regression. The Risø dataset was used for this comparison, because

both high and low fluxes were observed. Flux estimates of the two approaches matched reasonably well; significant differences were only observed at very high rates (Figure 4A and 4B). 85 % of the variance in N$_2$O fluxes from 3-min closure could be explained by fluxes from 60-min and 10-min closure (Figure 4B). The relatively high slope of 1.80 was mainly caused by three exceptionally high fluxes where the 60-min method considerably overestimated values of the 3-min method. Standard errors of N$_2$O fluxes from both 3-min and 60-min closure were extremely low, i.e. in the order of 0.2 % of

the fluxes (Figure 4C) with median values of 0.17 and 0.06 µg N m$^{-2}$ h$^{-1}$ and arithmetic means of 0.21 and 0.20 µg N m$^{-2}$ h$^{-1}$ for the 3 and 60-min closure flux estimates, respectively.

For better comparison with other studies, we also compared HMR-based fluxes with robust linearly calculated fluxes from our GC measurements when the full 60-min cycle was taken into account. A linear regression analysis (data not shown) resulted in a slope of 0.97 and an $R^2$ value of 0.86 under the high flux regime in Risø with the data set of robust linearly

calculated fluxes being the independent variable. The mean campaign flux value from HMR-based calculations was 22 % higher than the average campaign value of the robust linear method. The difference between the two methods was even higher under the low flux regime in Braunschweig. Slope and $R^2$ value of a linear regression analysis were 1.82 and 0.42, respectively. Despite the high slope value, the mean campaign value of the robust linear method only reached 51 % of the value obtained from the HMR method.

A further intriguing analysis shows that standard errors were found to be invariant on QCL sampling frequency (Figure 5). We simulated different sampling times ranging from one tenth of a second to 25.6 sec, which corresponds to a frequency of 0.0390625 Hz, by excluding the respective intervals from the original 10-Hz dataset. Results show that the median of the standard error of the fluxes remains stable over a wide range of measurement frequencies. At a frequency class of 0.15 and lower (3 boxes on the right-hand side of Figure 5), which corresponds to a sampling time of ~5 sec and higher, lower and

upper quartile values begin to deviate and the median changes slightly.

### 3.3 GC vs. QCL-based fluxes under low and high exchange regimes

Time series of $N_2O$ fluxes and their associated standard errors using both the GC and the 3-min QCL linear regression method during the Braunschweig and Risø campaigns are given in Figure 6. QCL fluxes in Braunschweig were on a constantly low level ranging between 2 and 16 $\mu g$ N $m^{-2}$ $h^{-1}$, whereas GC-based fluxes at the same site were scattered between $-13$ and 39 $\mu g$ N $m^{-2}$ $h^{-1}$. A linear regression revealed no significant relationship between GC and QCL fluxes with a very low coefficient of determination of 0.036 (Figure 7A). While standard errors of the QCL method were always below 0.6 $\mu g$ N $m^{-2}$ $h^{-1}$, values of the GC method were distributed between 0.5 and 22.0 $\mu g$ N $m^{-2}$ $h^{-1}$. Although higher variability and higher standard errors in GC-based fluxes were evident, mean $N_2O$ flux rates of the entire observation period were almost identical when comparing the two analyzer types. $6.42 \pm 5.98$ $\mu g$ N $m^{-2}$ $h^{-1}$ and $7.77 \pm 0.13$ $\mu g$ N $m^{-2}$ $h^{-1}$ were found for the GC and the QCL method, respectively.

Under the high exchange regime at Risø, $N_2O$ fluxes of the two analyzer types matched considerably better (Figure 6D). Although the willow field was already fertilized on DOY 99, $N_2O$ fluxes did not start to increase until DOY 105 when a sharp rise in air temperature was observed. GC-based fluxes were lower than QCL-based fluxes (slope=0.50) as in most cases a non-linear model could not be fitted with only four data points. A linear regression between GC and QCL fluxes revealed a coefficient of determination of 0.48 (Figure 7B). Standard errors of the QCL method were again extremely low, i.e. <1 % of the flux value and were always below 1.0 $\mu g$ N $m^{-2}$ $h^{-1}$, while those from the GC method were on average in the range of 5 to 10 % of the flux value. Parallel operation of both methods was conducted from DOY 105 to 108. During this period, the campaign means were $117.8 \pm 0.2$ and $77.4 \pm 8.2$ $\mu g$ N $m^{-2}$ $h^{-1}$ for the QCL and GC method, respectively.

As standard errors of QCL-based $N_2O$ fluxes were on a constantly low level, no dependency on flux value was observed in any of the campaigns (Figure 7). The same was evident for GC-based fluxes in Braunschweig. At Risø, however, a slight but non-significant tendency of higher standard errors at higher flux rates was found. Only 8 % of GC data from Braunschweig met the criteria for flux calculation using the HMR model. At Risø, 38 % of GC data allowed for HMR flux calculation indicating that higher exchange regimes favor the usage of an exponential model when using the GC method.

### 3.4 $N_2O$ uptake and diurnal variation

Neither at Risø nor during the Braunschweig campaign soil $N_2O$ uptake was observed when using QCL based measurements. In only very few cases (n=5) $c'(t)$ were initially found to be negative, however, these data, which exhibited abnormally high standard errors, were discarded due to mechanical malfunctioning of the chamber system as a result of non-closure caused by distorted guiding racks through very high wind speeds at Risø (*cf.* Section 3.3).

Regarding GC-based data in our study, 2 out of 37 fluxes in Risø were negative. Note that GC-based fluxes in Risø were only determined between DOY 105.5 and 108.5 when fluxes were elevated due to fertilizer application. In Braunschweig, however, nearly 25 %, i.e. 50 out of 201 flux rates from the GC setup were showing $N_2O$ uptake with only 3 of the 50 negative flux rates being significant ($p<0.05$, $p$-values not corrected for multiple testing).

An investigation of the diurnal variability of N$_2$O fluxes showed that during the Braunschweig campaign – although only small differences were observed – highest fluxes were found during midday and early afternoon (~8.7 µg N m$^{-2}$ h$^{-1}$), while lowest N$_2$O efflux was measured shortly before midnight and before sunrise (~7.2 and 7.3 µg N m$^{-2}$ h$^{-1}$, respectively; Figure 8), thereby following a commonly observed temperature-driven pattern (*cf.* Section 4.4). In Risø, however, we found lowest

5 fluxes of ~18.2 µg N m$^{-2}$ h$^{-1}$ at midday and highest fluxes when it was dark peaking before midnight at ~32.0 µg N m$^{-2}$ h$^{-1}$ (only data of DOY <105.5 and DOY >108.5 were taken to exclude fertilizer effects). Error bars in Figure 8 indicate the standard error of the mean from all flux values in each bin. Each bin contains fluxes from 3-hour periods, i.e. from 00:00 to 03:00, 03:00 to 06:00, 06:00 to 09:00, etc. The mean values in Figure 8 are plotted in the center of each bin. Fluxes were binned due to irregular starting times of new chamber cycles. In general, a new chamber cycle could be started each full

10 hour, but to get a more robust diurnal pattern, we decided to bin data in the above-mentioned 3-hour containers. While the diurnal variation of N$_2$O fluxes from the Risø campaign is significant (p-value = 0.0059), the diurnal variation found during the Braunschweig campaign is not as the difference between mean minimum and maximum values is lower than the upper flux detection limit of ~2.6 µg N m$^{-2}$ s$^{-1}$.

## 4 Discussion

### 4.1 The curvature parameter $\kappa$ as a chamber performance criteria

The high time resolution of QCL data allowed for a closer look at the shape of the concentration increase. The general form of the curve is determined by the rate of transport of a diffusing trace gas into the chamber headspace, which declines throughout deployment because any increase in the headspace concentration results in a corresponding decline in the vertical

concentration gradient driving that transport (Rolston, 1986; Hutchinson et al., 2000; Livingston et al., 2006). The change in the rate of transport is the initial curvature kappa, i.e. the second derivative of the concentration change at $t$=0.

The fact that extremely low negative $\kappa$ values between $-10^{-4}$ and $-10^{0}$ – indicating quasi linearity in $\partial c/\partial t$ – were almost exclusively found under low flux conditions, whereas fluxes >100 µg N m$^{-2}$ h$^{-1}$ were only observed when $\kappa$ was $< -10^{1}$ (Figure 3A) means that at higher fluxes the curvature in $c(t)$ is concave suggesting concentrations that tend to plateau over

25 time with the saturation effect becoming larger at higher flux rates. Near zero fluxes, however, corresponding to $\kappa$ values around zero, indicate no considerable changes in N$_2$O concentrations, thus, hardly any alteration of the vertical concentration gradient over time. Furthermore, closure time was found to have an impact on the magnitude of $\kappa$ (Figure 3B). Longer chamber deployment led to higher curvature in $c(t)$, which was expected as concentration gradients decline over time when a considerable flux is measured (*cf.* Hutchinson and Mosier, 1981; Livingston et al., 2006; Pedersen et al., 2010).

Our results imply that at low to moderately high flux rates <200 µg N m$^{-2}$ h$^{-1}$ (*cf.* Figure 4D) and/or short chamber closure, the slight non-linearity in concentration change when calculating fluxes is of minor importance and the application of linear

models is acceptable, particularly with regard to other commonly observed errors such as those originating from soil disturbance, chamber placement (Christiansen et al., 2011), temperature, pressure and humidity perturbations, etc. (Parkin and Venterea, 2010). At higher fluxes, however, significant curvature in $c(t)$ expressed by large negative $\kappa$ values will most likely lead to a substantial underestimation of fluxes when using linear regression instead of applying an exponential model

for flux calculation (*cf.* Matthias et al., 1978; Jury et al., 1982; Anthony et al., 1995; Kroon et al., 2008; Section 3.2). In principle, several other reasons making flux determination with linear or exponential models problematic, may technically be found. These are exponentially increasing $N_2O$ concentrations after chamber closure due to possible dispersion effects leading to biased analyzer readings when the elevated gas concentration is initially not uniformly mixed with the air inside the tubing, placement of the sample tube inlet at the top of the chamber lid leading to an establishment of a temporary

concentration gradient in a weakly mixed chamber atmosphere or an insufficient dimension of the pressure compensation tube leading to a push back of air into the uppermost soil layer in the moment when the chamber is set onto the lid. However, none of these were observed during our campaigns, thereby indicating a robust setup and chamber design for reliable $N_2O$ flux calculations.

We also investigated the possible effect of ambient wind speed and direction on concentration build up characteristics

(Figure 3C and 3D, respectively) as differences between turbulence conditions outside the chamber may possibly vary from those conditions inside the chamber under changing wind speed. Theoretically, pores in the uppermost soil layer might be ventilated under high wind speed when no chamber is in place, thus a close coupling of the flux to the atmosphere exists. Consequently, the establishment of a steady state flux may be more postponed under these high wind speed conditions once the chamber is put onto the soil frame. Such time delay caused by slow filling up of the previously ventilated pore space in

parallel to the diffusion into the chamber might in principle explain exponentially increasing concentrations. However, $\kappa$ values (Figure 3C and D) and fluxes (not shown) were independent of both wind speed and direction, which is a further indicator that the chosen chamber design and setup can be used over a wide range of environmental conditions and neither seem to affect concentration build up characteristics nor resulting flux magnitudes.

## 4.2 Closure time and measurement frequency – How long and how often is enough?

Reviewing past decades of field chamber measurements for studying soil-atmosphere exchange of $N_2O$, several challenges and shortcomings emerged such as limited number of replicates or disturbance of the soil micro-environment due to chamber coverage and soil collar insertion (e.g., Hutchinson and Mosier, 1981; Parkin and Venterea, 2010). One way of getting a higher temporal resolution, thereby a higher number of replicates and keeping soil disturbance as low as possible is to reduce the chamber closure period, which also is expected to decrease deviation from linearity in concentration increase.

The overestimation of the 60-min method compared to the 3-min method as shown in Figure 4B causing a relatively high slope of 1.80 was mainly caused by three exceptionally high fluxes. Beside any form of unintended interferences to the 'natural steady-state flux' like for example disturbances through macrofauna, fluctuating pump performance or analyzer

malfunctions due to internal re-calibration during chamber deployment, much higher 60-min-based HMR fluxes compared to 3-min-based linear fluxes may be observed when one of two following concentration increase patterns are observed.

> (1) Slow initial increase of concentrations followed by steeper rise after some minutes. Slope of the linear fit will then be much lower than the one from the HMR fit (linear fit at $t_0$).

> (2) Steady linear start of concentration increase followed by sudden relatively sharp bend with lower linear increase afterwards. HMR fit will also have a much steeper slope at $t_0$ than the linear fit, which will be on top of the data points for the first few minutes.

Red dots in Fig. 4B indicate situations similar to those described under (2) above. Recent work e.g. by Kroon et al. (2008) and Forbrich et al. (2010) demonstrated that emission estimates from closed chamber measurements were significantly
underestimated when using linear regression methods instead of the slope of an exponential function at the beginning of chamber closure. However, their linear regression models were applied to longer periods, i.e. to 10-min periods by Kroon et al. (2008) also using an Aerodyne QCL spectrometer and to 25-min periods by Forbrich et al. (2010) using a GC setup. Kroon et al. (2008) also showed that linear estimates differed by up to 60 % compared to those from exponential methods with a systematic error due to linear regression being in the same order as the estimated uncertainty due to temporal
variation.

As shown in Figure 4C, standard errors of $N_2O$ fluxes from both 3-min and 60-min closure were extremely low, i.e. in the order of 0.2 % of the fluxes with median values of 0.17 and 0.06 µg N m$^{-2}$ h$^{-1}$ and arithmetic means of 0.21 and 0.20 µg N m$^{-2}$ h$^{-1}$ for the 3 and 60-min closure flux estimates, respectively. In comparison, Cowan et al. (2014a) also find low flux uncertainty of less than 1 to 2 µg N m$^{-2}$ h$^{-1}$. This implies that limiting the chamber closure period to 3 minutes benefits in
two ways. On the one hand, the soil column of interest is less disturbed by shorter coverage and/or the number of replicates can be significantly increased. As these measurements are automated, no further manual work is required. On the other hand, standard errors of fluxes remain extremely low. However, it is recommended to extend the chamber closure period to a minimum of five and a maximum of 10 minutes as slightly delayed concentration increases under low flux regimes may occur (in ~5 % of the cases in our study) and would lead to an underestimation of 3-min linear fluxes (see Figure S1 in the
Supplement). We therefore recommend skipping the first 2 minutes of data to guarantee undisturbed conditions that might have been caused in the moment when the chamber is set on the soil collar. The 'dead time' of the system, i.e. the time that passes between the moment when an air sample leaves the chamber and the moment when it reaches the analyzer, was ~10 s – given a tube length of 10 m, a flow rate of 1 L min$^{-1}$, and an ID of the tube of 4.6 mm – and was already considered in the recommendation.

Standard errors of $N_2O$ fluxes were found to be invariant on QCL sampling frequency (Figure 5). The conclusion we can draw from this finding is that chamber operators – in case an analyzer with a precision like the QCL presented in this study is available – can reduce their sampling time down to 5 seconds without risking an increase of the standard error of the flux, which would still be on a much lower level than those obtained from GC measurements (*cf*. results in Section 3.2).

## 4.3 Differences between GC and QCL-based fluxes

Our comparison of GC vs. QCL fluxes revealed that despite much higher precision, robustness, and temporal resolution in QCL measurements, GC is still a useful method to determine the average campaign $N_2O$ soil efflux. Although single flux values particularly under low exchange regimes did not match well, campaign means and medians were similar to those obtained by the QCL method. Under high exchange regimes, however, flux patterns matched considerably better, but resulted in larger absolute errors when comparing the campaign average, thereby leading to systematic errors (in our case an underestimation) when using the GC method at high $N_2O$ fluxes for the assessment of N balances. However, given the fact that the bulk of the annual efflux occurs after management events at a relatively short time scale (Flechard et al., 2007; Skiba et al., 2013), usage of a GC-based system will be prone to large uncertainties (*cf.* Figure 7).

While only 8 % of GC data from the Braunschweig campaign met the criteria for flux calculation using the HMR model, 38 % of GC from the Risø data allowed for HMR flux calculation indicating that higher exchange regimes favor the usage of an exponential model when using the GC method. Similar findings (37 % allowance for non-linear model application) were reported by Petersen et al. (2011). Forbrich et al. (2010) found percentages of 13.6 %, 19.2 %, and 9.8 % of GC measurements on hummocks, lawns and flarks, respectively, which were best fitted with an exponential model. Their measurements, however, were made for methane fluxes and under an even larger $\partial c/\partial t$ range than was prevalent in our two campaigns. The fact that higher fluxes in our study were associated with lower standard errors and accepted HMR application is corresponding well with $\kappa$ findings in Section 3.1 indicating that higher curvature in $c(t)$ coincided with higher fluxes (Figure 3B).

In general, chamber architecture is essential for headspace concentration buildup patterns given certain enclosure times, activity levels and headspace mixing. Our new chamber system performed well during the two campaigns for both analyzer setups. Through its specific design with not only vertically but also horizontally moving chambers, it will keep the soil column under relatively natural conditions. The only problem emerged at Risø when the guiding racks were slightly distorted under high wind speed conditions, i.e. when half-hourly means of wind speed were higher than 6 m s$^{-1}$. However, this problem could easily be fixed by tightening the guy wires that are attached to the aluminum rack. Commonly observed winter problems such as unnatural accumulation of snow inside the chamber and rime ice formation on the guiding racks and soil frame were not tested within this study, but will likely affect the ease of operation during harsh winter conditions.

## 4.4 Enabling investigations of flux pattern characteristics

From an ecological point of view, QCL measurements offer a new opportunity for robust quantification of soil $N_2O$ consumption. As $N_2O$ uptake via denitrification exists in theory and could be shown under controlled lab conditions (e.g., Firestone and Davidson, 1989), it has been a major challenge to measure reliable fluxes in the field due to the fact that the magnitude of $N_2O$ uptake by soils is usually very low (Schlesinger, 2013), thereby problematic to be determined by GC measurements that are subjected to low signal to noise ratios (e.g., Brümmer et al., 2008).

Our QCL-based measurements under the given soil, temperature and moisture conditions at Risø and Braunschweig did not result in any soil $N_2O$ uptake fluxes. In the study by Cowan et al. (2014b), approx. 10% of their fluxes on grazed grassland and barley sites were negative. However, only 4 out of 115 negative fluxes were above the LoD of the method, which was estimated to be 4 µg N $m^{-2}$ $h^{-1}$, thus being similar to ours (cf. Table 2).

GC-based data in our study showed 2 out of 37 and 50 out of 201 negative fluxes in Risø and Braunschweig, respectively. In Risø, only 3 of the 50 negative flux rates were found to be significant ($p<0.05$, $p$-values not corrected for multiple testing), thus stressing the challenge of a robust determination of soil consumption of this important greenhouse gas when using the common vial-GC approach. Due to the fact that in this study no $N_2O$ soil uptake was found when using the much more reliable QCL setup, a further investigation of this topic on a variety of soil types under different land uses, plant

communities, and climatic conditions is highly desired.

Besides investigating possible $N_2O$ soil uptake, the QCL methodology offers the opportunity to study diurnal variability of $N_2O$ fluxes. In a recent study by Shurpali et al. (2016) it has been pointed out that neglecting diurnal variations leads to uncertainties in terrestrial $N_2O$ emission estimates and should therefore be carefully taken into account when calculating nitrogen budgets. Similar to our study (Figure 8), they found reversed diurnal patterns under differing flux magnitudes.

Intriguingly, when mean $N_2O$ fluxes were in a range between 12 and 35 µg N $m^{-2}$ $h^{-1}$, in both this study (Risø low flux regime) and Shurpali et al. (2016) highest fluxes were found during nighttime and lowest fluxes around midday. A contrasting diurnal pattern was observed when fluxes were lower than during the Risø campaign, i.e. in Braunschweig (7.2 to 8.7 µg N $m^{-2}$ $h^{-1}$) or much higher due to fertilizer application as in Shurpali et al. (2016) (230 to 475 µg N $m^{-2}$ $h^{-1}$). In the latter campaigns, mean $N_2O$ fluxes were highest at midday and lowest during nighttime, which corresponds to earlier

findings (*e.g.*, Christensen, 1983; Du et al., 2006; Parkin and Kaspar, 2006; Brümmer et al., 2008; Alves et al., 2012) where temperature was proved to be the main controlling factor for $N_2O$ soil-atmosphere exchange. Our study highlights that through its high time resolution QCL-based measurements will not only help enhance process understanding of $N_2O$ exchange by disentangling the strength of different drivers of $N_2O$ production like temperature, soil moisture, nitrogen availability, and microbial activity, but has also the potential to provide new insight into bidirectional exchange

characteristics of other trace gases such as $CH_4$, which can be sampled simultaneously with our chamber system depending on analyzer type used.

## 5 Conclusions

A new chamber system for automated measurements of soil-atmosphere trace gas exchange was developed. The system was tested for $N_2O$ flux detection in a conventional vial air sampling setup and with a directly connected QCL spectrometer

under low and high exchange regimes. Through its specific design, the system prevents measurement spots from unintended shading and minimizes disturbance of throughfall, thereby complying with high quality requirements of long-term observation studies and research infrastructures. Curvature in $\partial c/\partial t$ proved to be invariant with wind speed and direction.

High correlation (slope=0.99; $R^2$=0.93) was found when comparing linear vs. modified HMR flux calculation methods for $F_{N2O}$<200 µg N m$^{-2}$ h$^{-1}$. Intriguingly, mean campaign N$_2$O fluxes measured by GC and QCL of 6.42 and 7.77 µg N m$^{-2}$ h$^{-1}$, respectively, matched fairly well under low flux conditions, whereas under high flux conditions a significant deviation was observed (77.40 vs. 122.95 µg N m$^{-2}$ h$^{-1}$ from GC and QCL, respectively). While mean standard errors were in a range of 10 to 93 % of the N$_2$O flux in low to high exchange regimes when using the GC approach, extremely low values for standard errors of 0.2 to 1.7 % of the flux under different exchange conditions were found for QCL measurements. When a fast-response analyzer is available, we recommend reducing chamber closure time to a maximum of 10 minutes and applying linear regression to a 3-min data window with rejecting the first two minutes after closure and a measurement frequency of 0.2 Hz, i.e. a sampling output of every 5 seconds. With its high precision and temporal resolution, QCL technology provides furthermore a powerful tool to investigate highly debated topics such as diurnal flux variability and soil N$_2$O uptake.

**Acknowledgements**

This study was supported by the Thünen Institute of Climate-Smart Agriculture through the German Federal Ministry of Food and Agriculture (BMEL) as well as the Integrated Carbon Observation System (ICOS) infrastructure through the Federal Ministry of Research and Education (BMBF). Funding for CB and JJR by the BMBF Junior Research Group NITROSPHERE under support code FKZ 01LN1308A is greatly acknowledged. We highly appreciate logistical support during the Risø measurements, which were conducted in the framework of the InGOS project. The excellent introduction and valuable help on laser spectrometer operation and maintenance by David D. Nelson and Mark Zahniser is gratefully acknowledged. Many thanks are owed to Florian Hackelsperger from the experimental research station of the Institute of Animal Nutrition for technical support during the Braunschweig campaign as well as Kerstin Gilke and Andrea Oehns-Rittgerodt for N$_2$O analyses in the GC lab. Michel Bechtold is thanked for discussing and calculating dispersion estimates.

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

**Table 1. Supplementary information on field campaigns.**

|  | Braunschweig | Risø |
|---|---|---|
| **Coordinates** | 52°17'52''N, 10°26'36''E | 55°40'50''N, 12°06'05''E |
| **Start observation period** | Nov 13, 2012 | Apr 10, 2013 |
| **End observation period** | Dec 12, 2012 | Apr 24, 2013 |
| **Total GC flux rates (n)** | 201 | 37 |
| **Total QCL flux rates (n)** | 187 | 158 |
| **Land use** | Grassland | Willow field (harvested) |
| **Fertilization, date** | No fertilization | Apr 9, 2013 |
| **Fertilization, amount** | No fertilization | 120 kg N ha$^{-1}$ |
| **Fertilization, type** | No fertilization | Mineral (ammonium nitrate), N-P-K 21-3-10 |
| **Soil texture** | Silty sand | Sandy loam |
| **Soil type** | Cambisol | Luvisol |

**Table 2: Features of the chamber-analyzer system used in this study.**

| | GC[*] (model: Shimadzu GC-2014) | QCL[*] (model: Aerodyne Research Inc. mini-QCLAS) |
|---|---|---|
| No. of chambers | 3 | 3 |
| Chamber closure time | 60 min | 60 min<br>10 min (recommended) |
| Sampling frequency | every 20 min | 0.1 sec (max)<br>5 sec (recommended) |
| No. of concentration records per chamber run | 4 | 36000 in 60 min<br>6000 in 10 min |
| No. of chamber cycles per day | 24 (max) | 72 (recommended)<br>144 (max) |
| Maximum number of samples | 168 (depending on autosampler size) | Limited only by data storage capacity of QCL's computer or external hard drive |
| Lag time | (~10 sec) | ~10 sec |
| $N_2O$ flux detection limit ($\mu g\ N\ m^{-2}\ h^{-1}$) | 13.0 | 2.6 |
| Mean campaign $N_2O$ flux ($\mu g\ N\ m^{-2}\ h^{-1}$) | BS (pref.[1]): 6.42<br>Risø (pref.[1]): 77.40 | BS (lin.): 7.77<br>Risø (lin.[2]): 122.95 |
| Mean campaign SE of $N_2O$ fluxes ($\mu g\ N\ m^{-2}\ h^{-1}$) | BS (pref.[1]): 5.98<br>Risø (pref.[1]): 8.17 | BS (lin.): 0.13<br>Risø (lin.[2]): 0.21 |
| Median campaign $N_2O$ flux ($\mu g\ N\ m^{-2}\ h^{-1}$) | BS (pref.[1]): 5.15<br>Risø (pref.[1]): 64.80 | BS (lin.): 7.38<br>Risø (lin.[2]): 105.43 |
| Median campaign SE of $N_2O$ fluxes ($\mu g\ N\ m^{-2}\ h^{-1}$) | BS (pref.[1]): 5.04<br>Risø (pref.[1]): 4.72 | BS (lin.): 0.10<br>Risø (lin.[2]): 0.17 |
| Percentage of flux estimates where HMR could be fitted | BS: 8.5 %<br>Risø: 37.9 % | BS: 100 %<br>Risø: 100 % |

*GC – Gas chromatograph, QCL – Quantum cascade laser spectrometer, [1]preferred means non-linear HMR model was used if applicable, otherwise robust linear regression was taken, [2]mean/median of DOY 105.5 to 108.5 to make it comparable to GC data set*

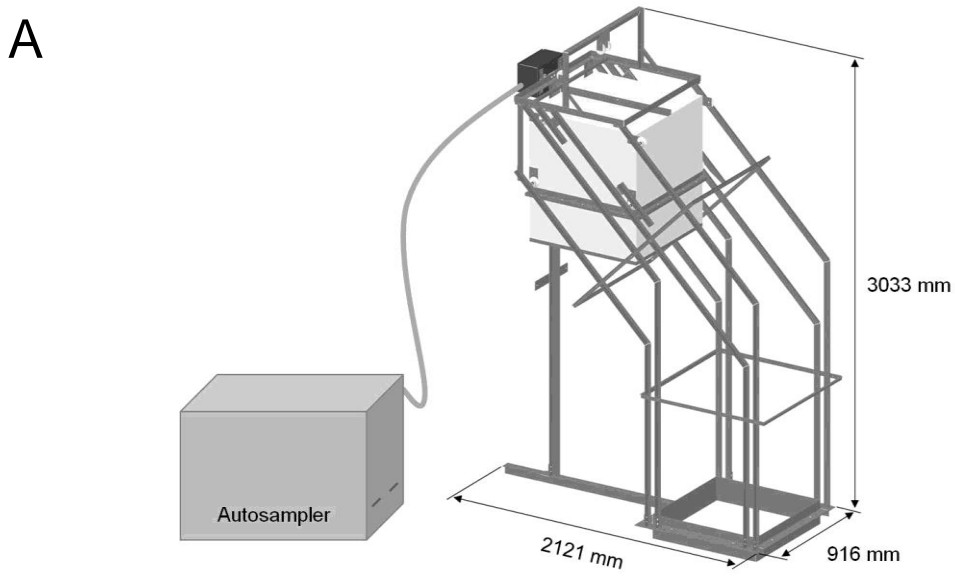

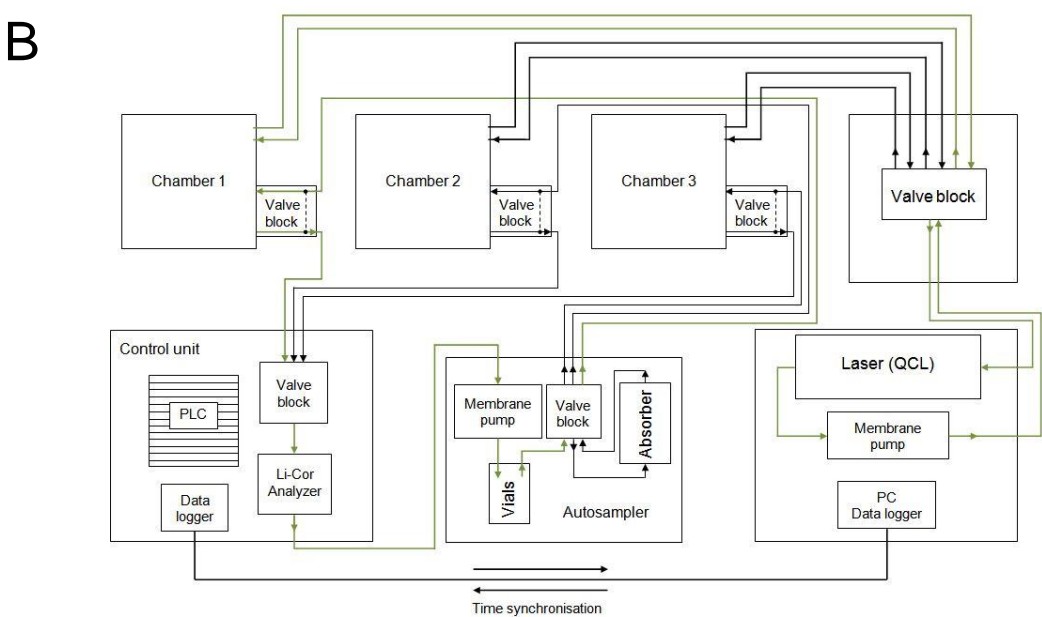

**Figure 1. Schematic diagrams of an automated chamber connected to an autosampler unit (A) and of the entire chamber system (B). Green lines indicate that Chamber 1 is currently in measurement mode. See text for detailed description.**

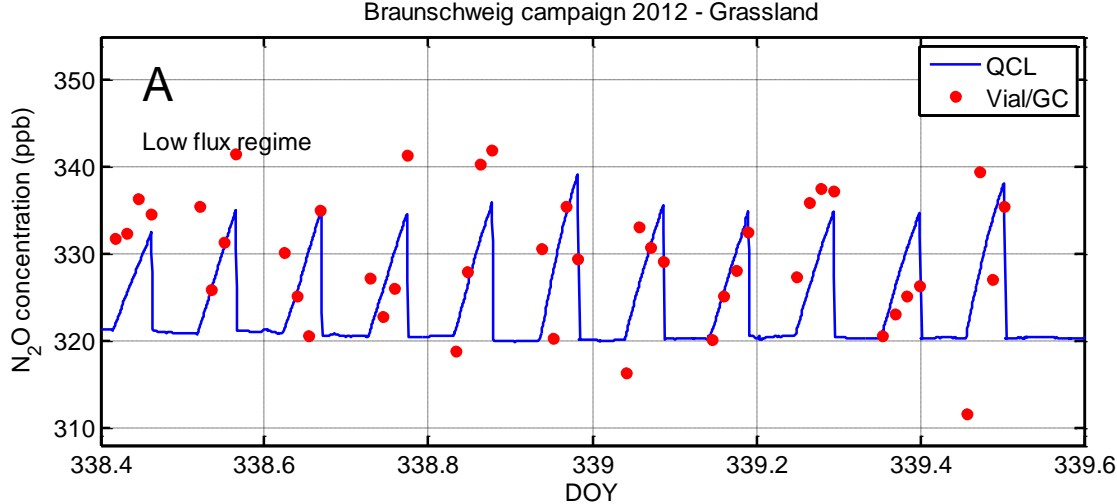

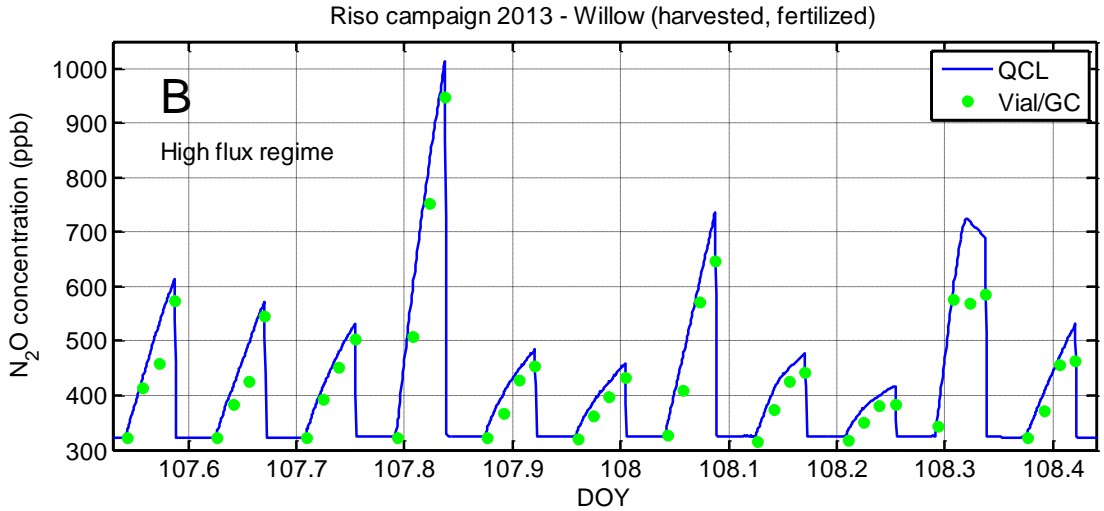

**Figure 2. Examples of time series of N₂O chamber concentrations during the Braunschweig (Panel A) and Risø campaign (Panel B). Chambers were periodically closed for 60 minutes. Vials were filled with sample air at $t_0$, $t_{20}$, $t_{40}$, and $t_{60}$. The QCL system was operated at a sampling frequency of 10 Hz; plotted are 1-min means.**

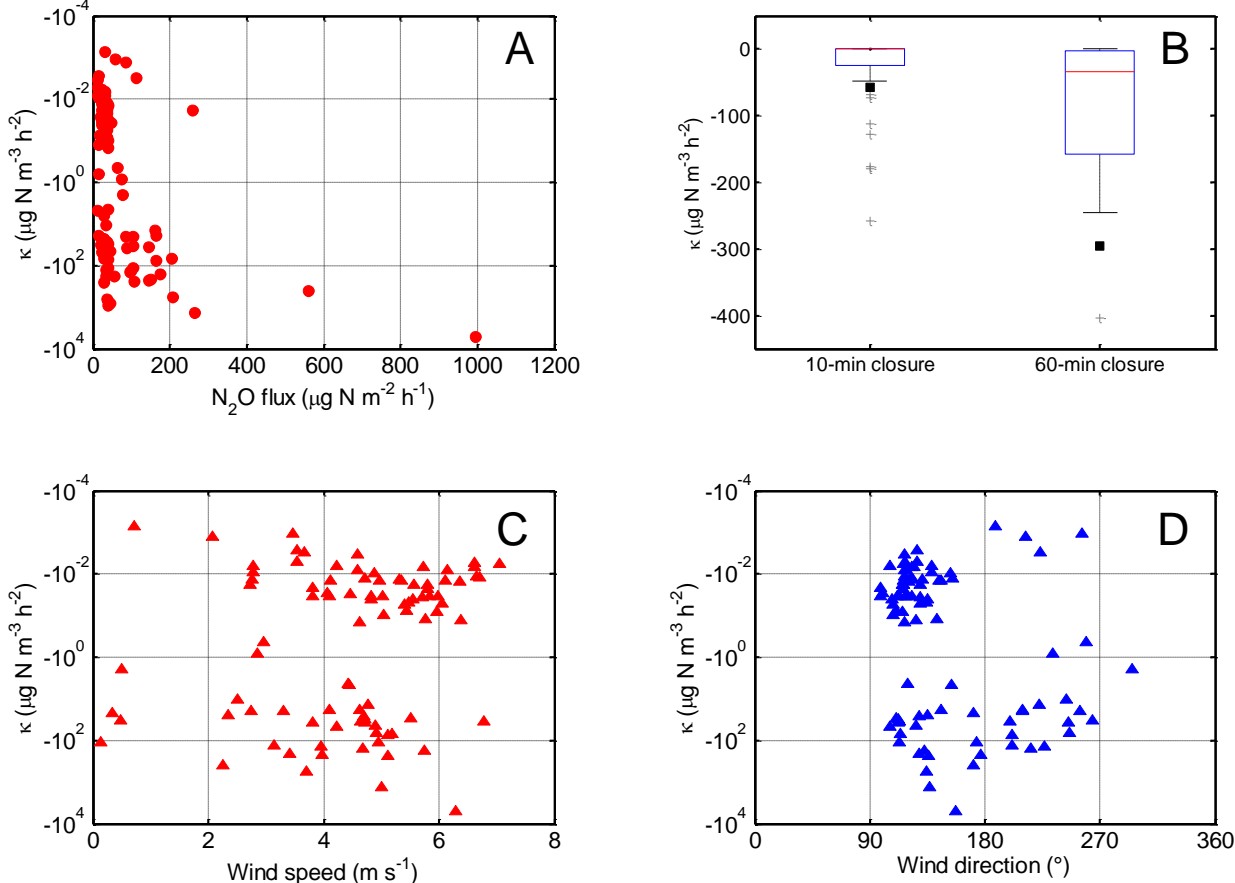

**Figure 3. Panels A: Relationship between $\kappa$ and N$_2$O fluxes calculated with an exponential model (see text for details). The parameter $\kappa$ indicates the curvature, i.e. the second derivative of the exponential model used for flux calculation. Negative $\kappa$ values correspond to concave functions, i.e. plateauing (saturating) N$_2$O concentration increases (*cf*. Figure 2). Panel B: Box plot of $\kappa$ values showing the difference between 10 and 60-min closure where black squares represent the arithmetic mean, red horizontal lines indicate the median, blue horizontal lines indicate lower and upper quartile values, black whiskers represent the interquartile range and outliers from this range are plotted as grey crosses. To ensure better readability, the y-axis is truncated at −450 μg N m$^{-3}$ h$^{-2}$. Thus, some outliers between −450 and −10$^4$ μg N m$^{-3}$ h$^{-2}$ are not shown. Panel C and D: Relationship between $\kappa$ and wind speed (C) as well as $\kappa$ and wind direction (D). All data are taken from the quantum cascade laser system operated during the Risø campaign. Chambers were closed for 10 minutes at DOY <105.5 and for 60 minutes at DOY >105.5.**

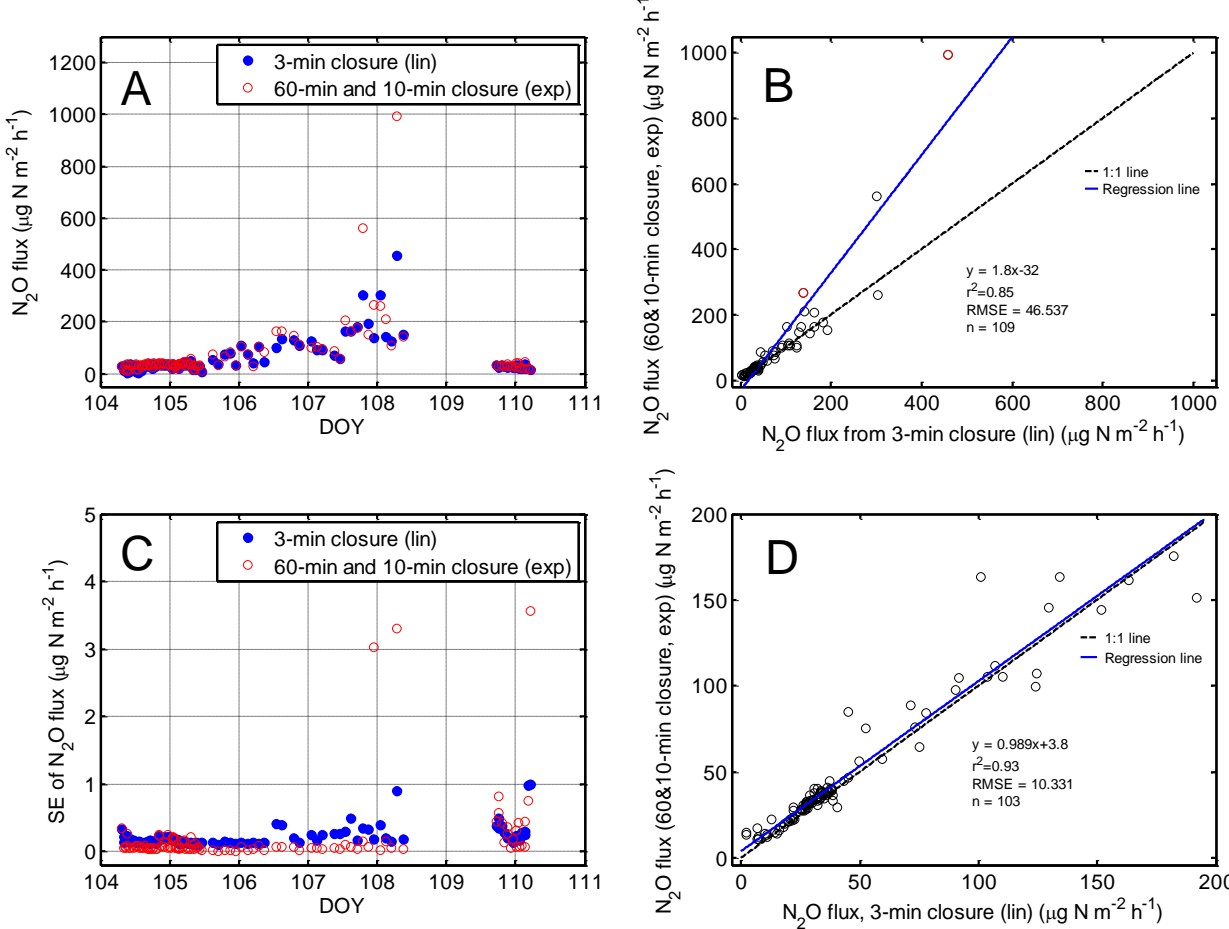

**Figure 4. Panel A: Comparison of N₂O fluxes measured on a harvested willow field during the Risø campaign by the QCL system based on a linear model using only the first three minutes of data after chamber closure (filled blue circles) and an exponential model (open red circles) (see text) using either the full 60 minutes (DOY 105.5 to DOY 108.5) or the full 10 minutes of data (DOY <105.5 and DOY >108.5). Panel B: Linear regression analysis of N₂O fluxes from the exponential vs. the linear model. Red circles indicate fluxes where the underlying concentration data showed an unusual pattern with a steady linear start followed by a sudden relatively sharp bend with lower linear increase afterwards (see Section 4.2 for details). Panel C: Standard errors of fluxes shown in Panel A. Panel D: Same as Panel B, but only for fluxes <200 µg N m⁻² h⁻¹ with adapted regression.**

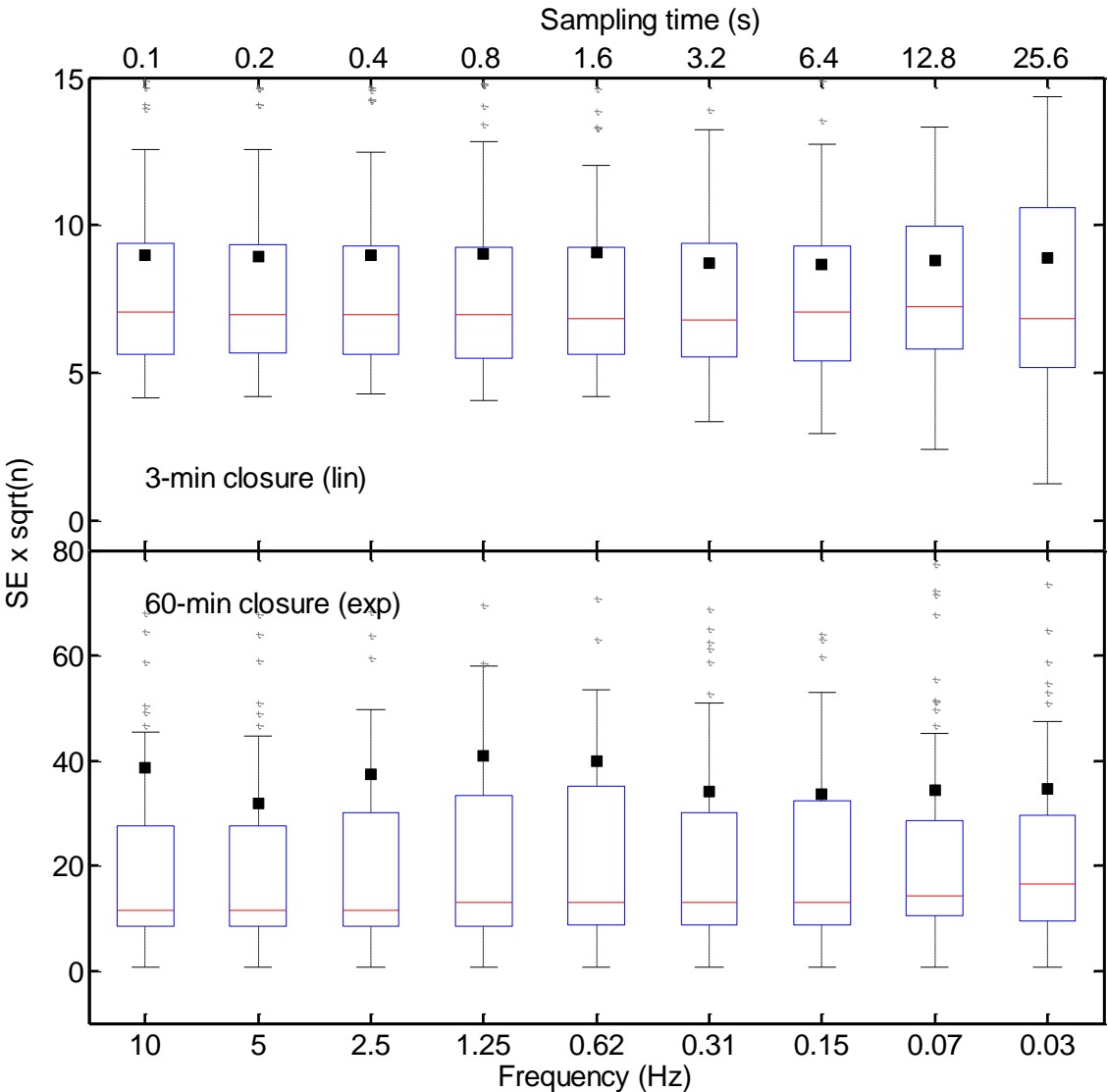

**Figure 5.** Boxplots of standard errors of $N_2O$ fluxes for different frequency classes and regression models used, i.e. linear regression with 3 minutes of data (upper panel) and the exponential HMR model with 60 minutes of data (lower panel). To avoid a pseudo-dependency on sample size, the standard errors (SE) were normalized by multiplication with $\sqrt{n}$. Black squares represent the arithmetic mean, red horizontal lines indicate the median, blue horizontal lines indicate lower and upper quartile values, black whiskers represent the interquartile range and outliers from this range are plotted as grey crosses.

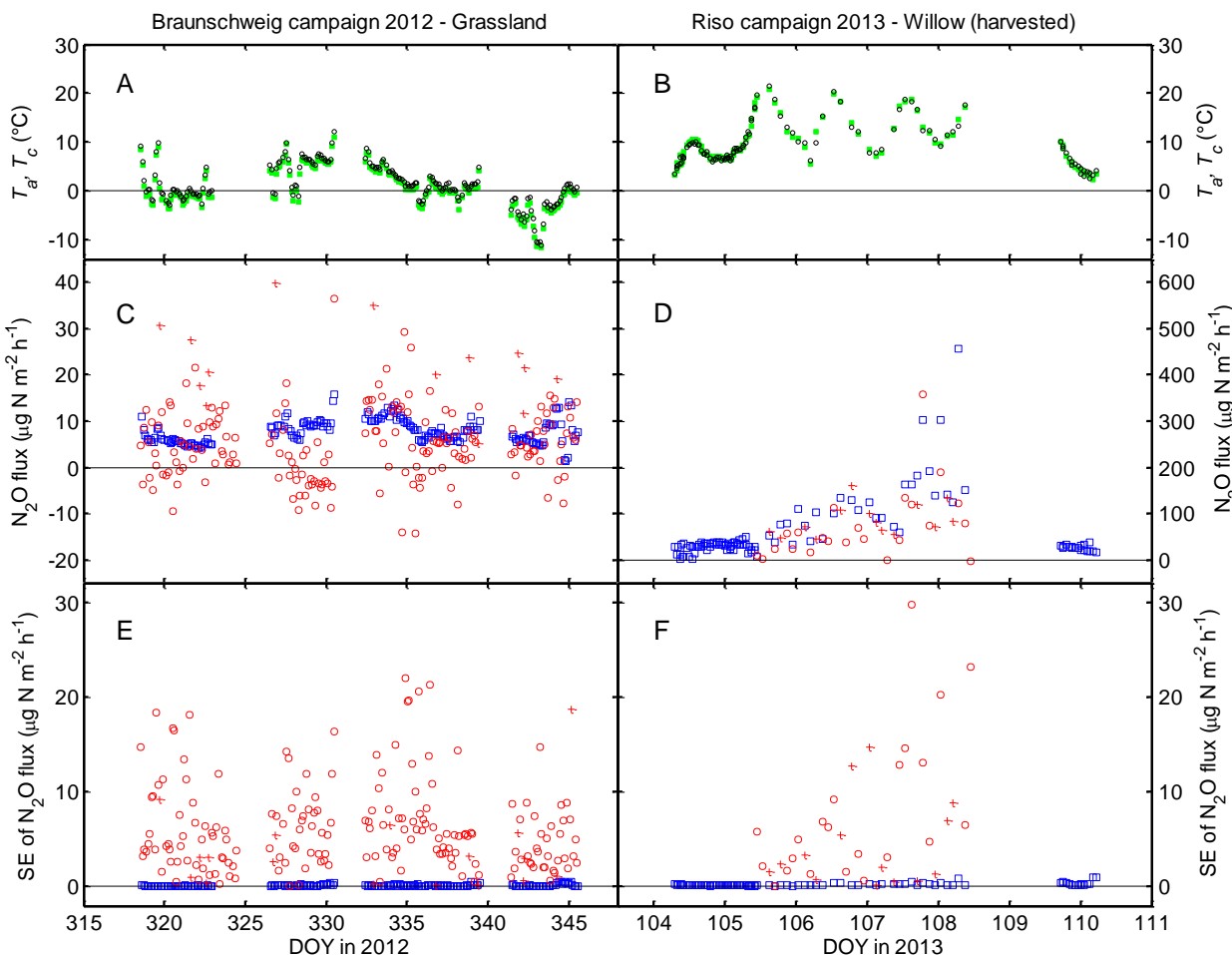

**Figure 6. Time series of air ($T_a$, green markers) and chamber temperatures ($T_c$, black markers)(panels A and B), N$_2$O fluxes and the respective standard errors of N$_2$O fluxes during the Braunschweig (panels C and E) and the Risø campaign (panels D and F). Blue markers indicate QCL data, red markers indicate GC data. Crosses are plotted for GC data when all criteria for flux calculation using the exponential HMR model were met (see text for details), otherwise circles are plotted indicating the usage of a linear model for flux calculation.**

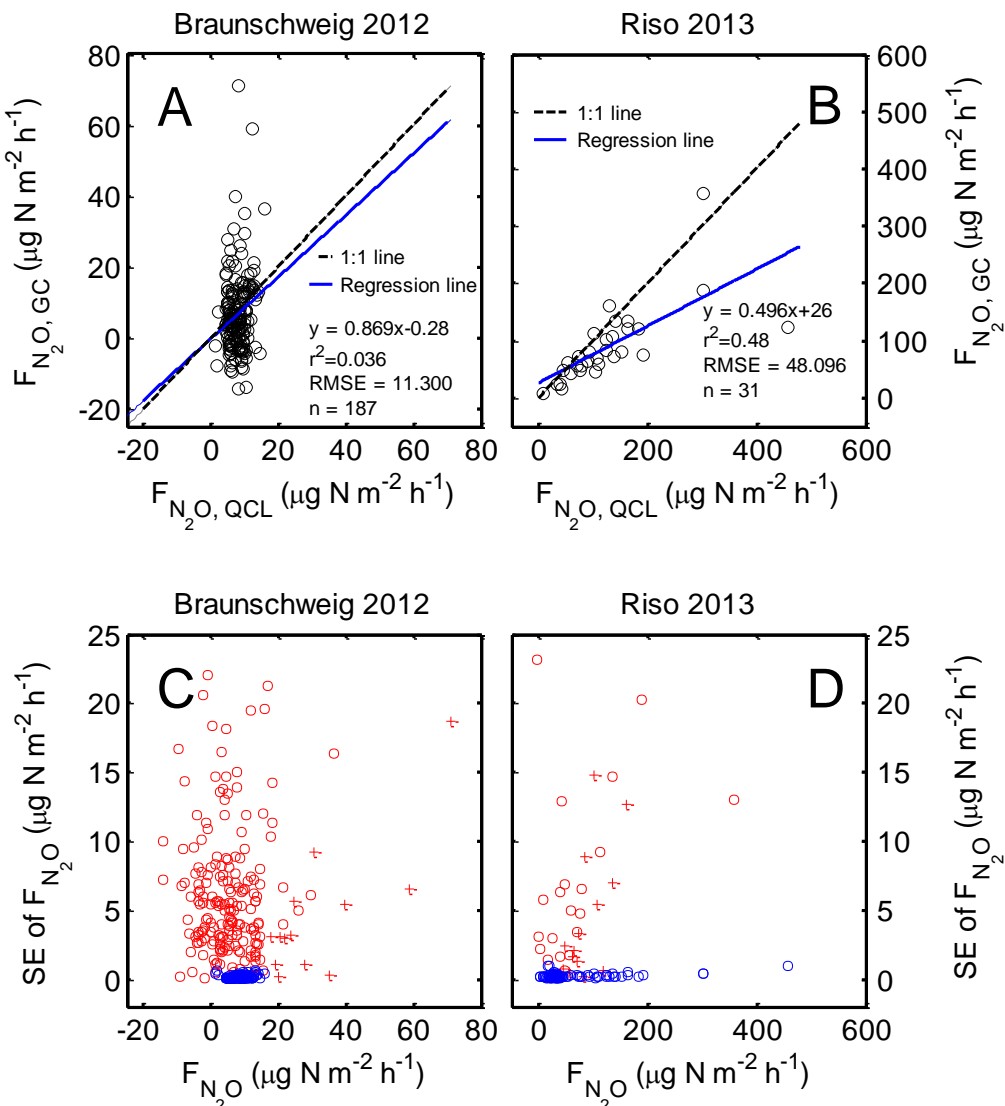

**Figure 7. Panels A and B: GC vs. QCL-based N₂O fluxes. Panels C and D: Relationships between standard errors (SE) of N₂O fluxes and the respective flux values. Blue markers indicate QCL data, which are all based on the 3-min linear calculation method. Red markers indicate GC data, which are based on the full 60-min data set. Crosses are plotted for GC data when all criteria for flux calculation using the exponential HMR model were met (see text for details), otherwise circles are plotted indicating the usage of a linear model for flux calculation.**

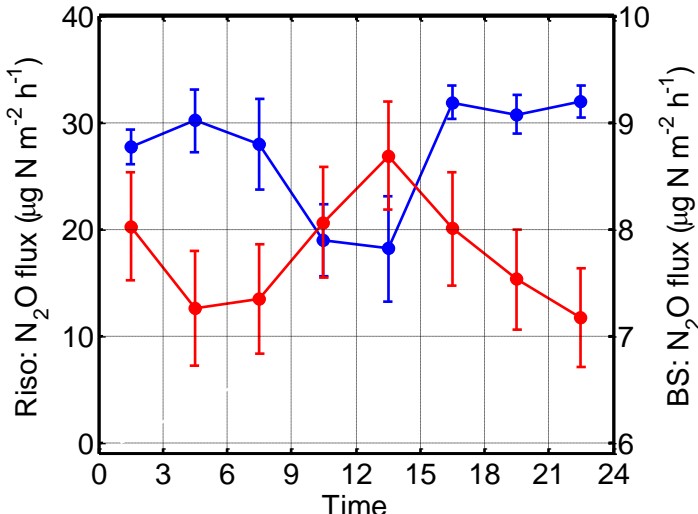

**Figure 8. Mean diurnal courses of N₂O fluxes derived from QCL flux measurements during the Risø (blue line) and Braunschweig (red line) campaign. To exclude fertilization effects in Risø, only data from the low flux period (DOY<105.5 and >108.5) were taken.**