# Peer review of "Gas chromatography vs. quantum cascade laser-based N2O flux measurements using a novel chamber design"

_Biogeosciences, 2016_

## Short Comment (SC1) · 8 Jul 2016

The manuscript "Gas chromatography vs. quantum cascade laser-based N2O flux measurements using a novel chamber design" by C. Brümmer et al,. is a methodical and well written study. The results show that modern quantum cascade lasers are able to out-perform aging GC methodology when it comes to N2O flux measurements from soils. The paper highlights some strengths of the new methodology and I believe that it should be published.

However, I do have some concerns with the cited literature. This is not the first study of its kind and the novelty of the setup could be questioned. An almost identical experiment was carried out in:

Cowan, N. J., Famulari, D., Levy, P. E., Anderson, M., Bell, M. J., Rees, R. M., Reay, D. S. and Skiba, U. M.: An improved method for measuring soil N2O fluxes using a quantum cascade laser with a dynamic chamber, Eur. J. Soil Sci., 65(5), 643–652, doi:10.1111/ejss.12168, 2014.

Both of these studies conclude very similar points and I believe that this paper should be cited in both the introduction and discussion part of the manuscript before publication. Further examples of this closed loop chamber methodology include:

Hensen, A., Groot, T.T., van den Bulk, W.C.M., Vermeulen, A.T., Olesen, J.E. & Schelde, K. 2006. Dairy farm CH4 and N2O emissions, from one square metre to the full farm scale. Agriculture, Ecosystems & Environment, 112, 146-152.

Laville, P., Lehuger, S., Loubet, B., Chaumartin, F. & Cellier, P. 2011. Effect of management, climate and soil conditions on N2O and NO emissions from an arable crop rotation using high temporal resolution measurements. Agricultural & Forest Meteorology, 151, 228-240.

I believe it would improve the manuscript to mention some of these papers, at least in the introduction section, if not also the discussion when comparing results. Another reference that is very relevant when investigating negative fluxes/instrumental detection limits is:

Cowan, N. J., Famulari, D., Levy, P. E., Anderson, M., Reay, D. S. and Skiba, U. M.: Investigating uptake of N2O in agricultural soils using a high-precision dynamic chamber method, Atmospheric Meas. Tech., 7(12), 4455–4462, doi:10.5194/amt-7-4455-2014, 2014.

The uncertainty cited in the manuscript for the fluxes measured using the QCL chamber method in the abstract was $\sim$0.1%; however, how can the authors be so sure of chamber volume? The uncertainty in flux is not the same as uncertainty in dc/dt. In the flux equation the uncertainty in the volume of the chamber is relative to the height

measurement, which on a uniform flat surface is negligible, but on a soil surface is more difficult to measure. Surely this uncertainty is at least 1% if not an order of magnitude greater, and so when propagated with uncertainty in dc/dt the flux uncertainty must also rise. See above references for examples.

There is no mention of a lag time between the instrument and chamber. It is suggested the first two minutes of measurement data are removed to avoid artifacts from soil disturbance. Does this also cover the time it takes for the gas to circulate fully between chamber to instrument and back to chamber again. If not then this "dead time" should be extended until the closed loop completes one full circulation to ensure mixing of the air within the tubing and chamber.

Uncertainties in comparisons of fluxes seem relatively low. Have you used standard errors in these comparisons? Would 95 % confidence intervals not be more relevant when comparing measurements known to have such large spatial and temporal variability?

A mobile field scale experiment was carried out using a similar methodology. It may not fit with this specific manuscript, but I include it for the author's interest.

Cowan, N. J., Norman, P., Famulari, D., Levy, P. E., Reay, D. S. and Skiba, U. M.: Spatial variability and hotspots of soil N2O fluxes from intensively grazed grassland, Biogeosciences, 12(5), 1585–1596, doi:10.5194/bg-12-1585-2015, 2015.

---

## Author Comment (AC1) · 30 Jul 2016

**Response to interactive comment (SC1) on 'Gas chromatography vs. quantum cascade laser-based N$_2$O flux measurements using a novel chamber design'**

*[SC] The manuscript "Gas chromatography vs. quantum cascade laser-based N2O flux measurements using a novel chamber design" by C. Brümmer et al,. is a methodical and well written study. The results show that modern quantum cascade lasers are able to out-perform aging GC methodology when it comes to N2O flux measurements from soils. The paper highlights some strengths of the new methodology and I believe that it should be published.*

*However, I do have some concerns with the cited literature. This is not the first study of its kind and the novelty of the setup could be questioned. An almost identical experiment was carried out in:*

*Cowan, N. J., Famulari, D., Levy, P. E., Anderson, M., Bell, M. J., Rees, R. M., Reay, D. S. and Skiba, U. M.: An improved method for measuring soil N2O fluxes using a quantum cascade laser with a dynamic chamber, Eur. J. Soil Sci., 65(5), 643–652, doi:10.1111/ejss.12168, 2014.*

*Both of these studies conclude very similar points and I believe that this paper should be cited in both the introduction and discussion part of the manuscript before publication.*

[AC] We highly appreciate the advice to cite the Cowan et al. (2014) paper, which in fact deals with very similar points and we fully agree to mention it in both introduction and discussion. We will add an extra line on Page 3, Line 10 stressing that first examples of high-resolution chamber measurements have been successfully carried out by Cowan et al. (2014a; 2014b) as well as by other teams like Hensen et al. (2006), Laville et al. (2011), Savage et al. (2014) or Sakabe et al. (2015). Further, we will compare findings of standard error calculation from our manuscript to the Cowan et al. (2014) paper in the discussion section (Page 8, Line 28). For differences in the uncertainty estimation between the two studies, please see respective comment below.

Regarding the comment '…the novelty of the setup could be questioned.', we like to clarify that at no point in our manuscript we claim that the measurement setup or any of the flux calculation methodologies have not been shown elsewhere before. The term 'novelty' (or 'novel') exclusively refers to the chamber design, which has so far not been presented in peer-reviewed literature. The aim of our paper is to highlight differences between GC and QCL setup using a 'novel' chamber system and to determine associated differences when applying linear vs. non-linear models to these datasets.

*[SC] Further examples of this closed loop chamber methodology include:*

*Hensen, A., Groot, T.T., van den Bulk, W.C.M., Vermeulen, A.T., Olesen, J.E. & Schelde, K. 2006. Dairy farm CH4 and N2O emissions, from one square metre to the full farm scale. Agriculture, Ecosystems & Environment, 112, 146-152.*
*Laville, P., Lehuger, S., Loubet, B., Chaumartin, F. & Cellier, P. 2011. Effect of management, climate and soil conditions on N2O and NO emissions from an arable crop rotation using high temporal resolution measurements. Agricultural & Forest Meteorology, 151, 228-240.*

*I believe it would improve the manuscript to mention some of these papers, at least in the introduction section, if not also the discussion when comparing results.*

[AC] We agree that it would be helpful to refer to some earlier studies where high-resolution measurements were used. We will include Hensen et al. (2006) and Laville et al. (2011) in the introduction section (see comment above).

*[SC] Another reference that is very relevant when investigating negative fluxes/instrumental detection limits is:*

*Cowan, N. J., Famulari, D., Levy, P. E., Anderson, M., Reay, D. S. and Skiba, U. M.: Investigating uptake of N2O in agricultural soils using a high-precision dynamic chamber method, Atmospheric Meas. Tech., 7(12), 4455–4462, doi:10.5194/amt-7-4455-2014, 2014.*

[AC] This is a very good hint and an excellent paper regarding the discussion about $N_2O$ uptake by soils. We will add a few lines on Page 11, Line 3 highlighting their observation of approx. 10 % negative fluxes with only 4 out of 115 measured flux rates being above the limit of detection (LOD) and will compare these findings to our results.
Regarding the detection limit of our QCL and GC setups, LOD could be estimated using our campaign data assuming stationary conditions during the low flux campaign in Braunschweig. Taking the whole campaign into account, the calculated standard deviations were 2.5 µg m$^{-2}$ h$^{-1}$ and 7.5 µg m$^{-2}$ h$^{-1}$ for QCL and GC measurements, respectively. Thus, the resulting 2-$\sigma$ uncertainty range for QCL was 5.0 µg m$^{-2}$ h$^{-1}$ and for GC 15.0 µg m$^{-2}$ h$^{-1}$. If only the first quarter of the Braunschweig campaign data are taken, i.e. a period where environmental conditions were less variable than over the whole campaign, the calculated standard deviations were 1.3 µg m$^{-2}$ h$^{-1}$ and 6.5 µg m$^{-2}$ h$^{-1}$ for QCL and GC measurements, respectively. Thus, the resulting 2-$\sigma$ uncertainty range for QCL was 2.6 µg m$^{-2}$ h$^{-1}$ and for GC 13.0 µg m$^{-2}$ h$^{-1}$. These estimates can be regarded as an upper flux detection limit. A supposable lower flux detection limit solely depends on the sensitivity of the analyzers. Precision of the QCL is 0.03 and 0.01 ppb when averaging over 1 and 60 s, respectively. We will add a few lines at the end of chapter 2.2 on Page 5, Line 20 to provide this information.

*[SC] The uncertainty cited in the manuscript for the fluxes measured using the QCL chamber method in the abstract was ~0.1%; however, how can the authors be so sure of chamber volume? The uncertainty in flux is not the same as uncertainty in dc/dt. In the flux equation the uncertainty in the volume of the chamber is relative to the height measurement, which on a uniform flat surface is negligible, but on a soil surface is more difficult to measure. Surely this uncertainty is at least 1% if not an order of magnitude greater, and so when propagated with uncertainty in dc/dt the flux uncertainty must also rise. See above references for examples.*

[AC] This must be a misunderstanding and will be clarified in the manuscript. We do not investigate uncertainty that is directly related to observational errors or any kind of measurement or setup issues, e.g. changes in flow rate, temperature sensitivity of the QCL, pump performance, changes in chamber volume due to rough soil surfaces or plants in the chamber, etc. As it is one of the main aims of the paper, we just simply look at a comparison of standard errors associated with the flux calculation method for GC and QCL. This approach ensures quantitative comparability between linear vs. non-linear regression models on the one hand and GC and QCL on the other hand. (Surely the regression uncertainties are indirectly influenced by observational errors). Further, if we would include the error of the effective chamber height, the uncertainty for both (GC and QCL) would be affected in the same way. We appreciate the comment that the reader might be misled. We will add a sentence at the end of Section 2.3, Page 6, Line 17 for clarification.

*[SC] There is no mention of a lag time between the instrument and chamber. It is suggested the first two minutes of measurement data are removed to avoid artifacts from soil disturbance. Does this also cover the time it takes for the gas to circulate fully between chamber to instrument and back to chamber again. If not then this "dead time" should be extended until the closed loop completes one full circulation to ensure mixing of the air within the tubing and chamber.*

[AC] Good point. It wasn't mentioned that the 'dead time' is already included in our suggestion to remove the first two minutes of data to avoid artifacts from soil disturbance. 'Dead time' within the QCL setup was ~10 s given a tube length of 10 m, a flow rate of 1 L $min^{-1}$, and an inner diameter of 4.6 mm (note that ID of the sample tube within the QCL setup was different from ID of the sample tube within the GC setup; cf. Page 4, Line 16). We will add 'In our case, the 'dead time' of the QCL measurement system, i.e. the time that passed between an air sample leaving the chamber and entering the analyzer, was ~10 s given a tube length of 10 m, a flow rate of 1 L $min^{-1}$, and an inner diameter of 4.6 mm.' on Page 9, Line 2.

*[SC] Uncertainties in comparisons of fluxes seem relatively low. Have you used standard errors in these comparisons? Would 95 % confidence intervals not be more relevant when comparing measurements known to have such large spatial and temporal variability?*

[AC] Yes, we used standard errors of the regression model as mentioned several times in the text and as shown in Figures 4 to 7. Although in general $N_2O$ flux measurements are known to feature large spatial and temporal variability, we compare in our study fluxes measured simultaneously in the same chamber. The aim is a quantitative comparison of different flux calculation methods and not the investigation of temporal or spatial heterogeneities. Therefore we explicitly want to point out the statistical error associated with the linear or non-linear regression model. For sparse GC sample data the standard error of the regression is more meaningful. A rough estimation of the 95 % confidence intervals is two times the standard error. Any other more complex statistics do not make much sense for only 4 data points.

*[SC] A mobile field scale experiment was carried out using a similar methodology. It may not fit with this specific manuscript, but I include it for the author's interest.*

*Cowan, N. J., Norman, P., Famulari, D., Levy, P. E., Reay, D. S. and Skiba, U. M.: Spatial variability and hotspots of soil N2O fluxes from intensively grazed grassland, Biogeosciences, 12(5), 1585–1596, doi:10.5194/bg-12-1585-2015, 2015.*

[AC] Thank you for mentioning this paper. However, as the same analytical devices are used as in Cowan et al. (2014a; 2014b) and it mainly deals with spatial variability, we will not include it in our manuscript.

References:

Sakabe, A., Kosugi, Y., Takahashi, K., Itoh, M., Kanazawa, A., Makita, N., Ataka, M.: One year of continuous measurements of soil $CH_4$ and $CO_2$ fluxes in a Japanese cypress forest: Temporal and spatial variations associated with Asian monsoon rainfall, J. Geophys. Res. – Biogeosciences, 120(4), 585–599, 2015.

Savage, K., Phillips, R., and Davidson, E.: High temporal frequency measurements of greenhouse gas emissions from soils, Biogeosciences, 11, 2709–2720, 2014.

---

## Referee Comment (RC1) · Anonymous Referee #1 · 2 Aug 2016

Brümmer et al. present a study analyzing linear and non-linear flux calculation methods under high and low flux rates of nitrous oxide and different scenarios of closure time. They use both traditional gas chromatography (GC) (low sampling number during closure time) and high-resolution quantum cascade laser (QCL) sampling. They find that non linear concentration changes are more clearly detectable during high emission scenarios and long chamber closure. Shortening of closure time results in a reasonable agreement between linear (3min) and non-linear (60min) flux estimates, but can only be applied when using the QCL set up. While under low flux conditions, GC measurements result in more scattered flux estimates, in both campaigns mean flux estimates of GC and QCL agreed well. Rare negative fluxes detected by GC mea-

surements seem to be arbitrary and not caused by actual N2O uptake. The paper is well written and a good fit for the journal. However, I could not help thinking that most of the results were as to be expected from literature and not 'radically' new. Ultimately, the high temporal resolution of measurements possible with QCL (which do provide more sophisticated ways of flux data processing) and the fact that the concentration measurements are instantaneous, make these measurements desirable for exactly the long-term applications, the authors are suggesting. It is not clear to me, what the accessibility of the described instrumentation is. Is there a plan to make it available for other users, i.e. to 'rent' it out or to make it available within the ICOS project? If that is the case, it should be pointed out more clearly. Overall, the most interesting aspect to me is the possibility to study ecological processes in a new way, as shown for the possible net N2O uptake and diurnal variability in emission rates. Interestingly, the study they compare their results to (Shurpali et al. 2016) is mostly an eddy covariance study. It would be interesting if the authors could comment on possible advantages of this automatic chamber against eddy covariance and whether other gases can be sampled in parallel to N2O (I am thinking mostly of CO2, considering the possible coupling of plant activity and N2O emission rates).

---

## Referee Comment (RC2) · Anonymous Referee #2 · 1 Sep 2016

**Brummer et al: Gas chromatography vs quantum cascade laser-based N2O flux measurements using a novel chamber design**

The authors have tested the performance of a QCL analyzer connected to a new automated chamber, against a "conventional" GC + automated gas sampling unit system. Data from QCL system were used to observe the non-linearity in the concentration increase during the chamber closure. Based on two short campaigns, the paper gives recommendations how should the measurements, data screening and flux calculations be done. The new chamber design is interesting and the system coupled to QCL seems to be fluently producing nice data. Papers presenting new chamber designs, are always welcome, particularly if they can provide generalizations and recommendations which are useful for other chamber operators. The paper is fluently written, and the observation of different patterns in diurnal cycle is interesting and important. However, there are several deficiencies and pitfalls in the data treatment and the argumentation which need revision. The presentation quality would benefit from separating the results and discussion.

First, the performance of the GC sampling system makes me wonder whether the comparison of two systems is meaningful. Before making any comparisons, the authors should find out the reason for the bad performance of the GC. Secondly, there are several conclusions in the paper which are just qualitative, and as such they are vague and are not supported by the presented data. Third, I share the worry of the first reviewer that most of the results shown here are already well known. For example, it has been reported already in numerous papers that the curvature in concentration increase is higher with longer closure time, and that using the linear calculation instead of non-linear can result in great underestimate in flux rate. Instead of reporting curvature, it would be more useful to quantify what is the limit of curvature after which the authors recommend the use of non-linear fitting method. Also, it would have been interesting to learn more about the advantages and possible problems in the "novel" chamber design. In general, the paper could be more valuable would it provide more quantitative information and recommend some general tests which each chamber operator should run to ensure adequate data quality. It is also a bit questionable if the paper with such a short piece of data (25 + 6 days) is enough the draw firm conclusions.

See more comments below.

**MORE DETAILED COMMENTS:**

What is the reason for the very bad performance of the GC system? On p5 lines 7-10 it is said that the system was checked against ten samples of ambient air, and only if the CV falls <3%, the data is acceptable. Is this CV limit of 3% really acceptable for a GC system? From Figs. 2a and 7a it seems clear that the GC is not able to resolve concentration increases for fluxes < 20 µgN m-2 h-1. At least to my knowledge, much lower fluxes analyzed with the GC have been reliably reported. I think that a comparison between QCL and GC is not really meaningful if GC is not able to measure these "small" fluxes of N2O. However, Fig. 2a makes me doubt, whether the problem is in the autosampler, and not in the GC detection limit? In some cases the GC can quite perfectly detect a concentration increase of about 12 ppb's similarly to the QCL (second measurement of DOY 339), but during many other closures the data seems arbitrary. What is the reason for that?

There are conclusions in the paper which are not supported by the presented data. For example: p.1 line 30: "new chamber design reduces the disturbance of the soil". There was nothing on that in the results. What are the possible disturbances? How can you detect that those can be omitted by your system? Or: lines 25-26: GC was found to be a useful method to determine N2O fluxes at longer time scale". Where is the data to prove

such a conclusion? There were no budgets calculated. What happens with the low fluxes, how can you reliably determine budget if you cannot detect the flux? Or p.8 line 31 forward: how do you justify the recommendation of removing the first 2 minutes of data? Or "1 to 5 s frequency was sufficient to keep SE on much lower level than in fluxes determined by the GC method" (p. 9). "sufficient" was not defined here. How do you justify the limit of using only the first 10 min of the data? Please give some argument based on the data, not just the feeling that this is good.

Many of the conclusions of the paper follow those observed in previous studies and are already well known. The one exception is the data in Fig. 8 showing the diurnal variation in N2O flux, as this phenomenon is not much studied. In addition to such data providing information on GHG formation processes, the value of the paper could have been in showing how exactly this chamber system works and what are the special and quantified  conditions needed to run the system and to screen the data in order to provide reliable flux data.

I strongly recommend to separate the results and discussion parts; presently it is difficult to follow the storyline.

**METHODS**

P.4 L.3 Why "semi-automatic"?

P.4 L.6-7 A volume of L x W x H does not result in 0.33 m3

**Chapter 2.3:**

p.5 L. 26: what are the conditions when HMR function cannot be fitted? ; L27, what is Akaike information criterion, please open this a bit, although there is the reference, the reader should get some kind of an idea just by reading the text here.

Equations 1-3 and the text related to them: add units.

**RESULTS**

P.7 L.7: "low negative k values": care should be taken to express the relations between negative and more negative values. Perhaps more clear to speak about absolute values when comparing these.

P.7. L 10-11 "Near zero fluxes indicate no considerable changes in N2O concentration". Isn't this self-evident without any measurements? Also, what is "considerable change in N2O concentration"? Do you mean significant? If there's no significant increase in concentration, there is no flux, true? Remove or reword the sentence. The whole chapter (Lines 7-14) seem quite self-evident, as the authors hint in the last sentence of the chapter. From Line 15 onwards you say that application of linear model is acceptable in some cases. However, no quantification, i.e. limit below which this is acceptable, is given. I also do not understand how do you draw this conclusion from the results on Lines 7-14.

P.7. L.23: what is meant with dispersion here?

P.7 L. 29-30 "…outside the chamber and inside chamber conditions…" please reword

P.7 L. 30-31 "…coupling of the flux under ambient conditions…" I do not understand this sentence, please reword

P.7 L.28 → What about the impact of the fan speed on curvature? Soil pores may be ventilated also by the fan (see for example Lai et al. 2012, BG).

**Chapter 3.2**

Might be good to start with your own results, not with the literature review. For clarity, I strongly recommend separating results and discussion.

P.8 L. 15 onwards: Figure 4B indicates that actually the 3-min/lin method produces higher fluxes than the 60 min/exp method in the lower flux regime (below 200 µg N m-2h-1). When taking into account also the higher fluxes (n=6), the relationship changes so that these 6 data points make a very strong impact, as the authors already discuss. Even though this is the case, the discussion here emphasizes continuously how the linear fluxes are smaller than exponential, although the results shown support this observation only for the few high flux points. This makes me to doubt how one can make generalizations about the validity of these two methods. I am missing discussion which tries to find explanation for the higher fluxes with 3-min/lin method. Is it so that the data set should be split, or is it far too small to make generalizations?

Would be also interesting to see, what happens to the SE/RMSE or similar, when apparently low fluxes are calculated with the exponential method. Is there perhaps a risk of higher error /noisy flux data? Would it be possible to find a flux rate below which the linear method is working more reliably than the exponential?

L. 25 I do not understand this sentence

P.8 L.31 onwards, continuing on P.9: Here you give important recommendations, but show no data. Also the reference to Section 3.1 is strange, as I do not find anything about the delayed concentration increase in 3.1. This data should be definitely shown if such recommendations are given. You should justify the removal of the first 2 minutes of data: why exactly 2 minutes?

P.8 L3. "…we also compared HMR-based fluxes **from QCL?** with robust linearly calculated…". How does this vary from that in Fig 7 upper right panel?

A general comment/hint: there are many different comparisons with different analyzers, calculation methods and closure times, in which partly different data sets have been used (low and/or high flux) and it is not easy to follow how do all these small experiments differ from each other or support each other. A separate result section with subsections dedicated to each of these questions might help in that. Now there is lot of text (e.g. Chapter 3.2) and it is difficult to follow the argumentation on logics of the text. Also a clearer division into paragraphs would help the reader. And, as already pointed out, division into results and discussion is needed.

P.9 L. 3-10: In which figure are these shown? Are the slope of 0.97 (lin fluxes are independent) and the HMR fluxes being 22% higher in conflict with each other? How is it possible that now the linear and HMR based fluxes estimated from 60-min data are almost identical (slope=0.97), while earlier you have stated that linear method underestimates the fluxes?

P.9 L 11-18: How were the standard errors calculated?

This section and Figure 5: I think that SE is not an appropriate quantity when estimating the "sufficient" frequency of concentration data. By definition, SE is related to (the root square of) the number of observations. It is therefore evident that if you decrease the frequency and the number of data, you increase the SE. In a case where the random error of the concentration measurement during the chamber closure is constant, the SE will

anyway increase in case the number of observations decreases, whereas the NRMSE, or the error in the flux will not increase. Therefore a better quantity to estimate the error related to the frequency of concentration data is RMSE (or NRMSE).

Your argument "..sampling times between 1 and 5 sec are sufficient to keep SE of fluxes on a much lower level…" is vague. How do you justify that exactly the 1-5 sec limit is sufficient? What means sufficient? How much is "much lower level"? Please quantify and justify this with an appropriate and objective criteria.

P.9 L.12: "..to approx.. one minute,…" isn't it approx. half a minute (25.6 sec)?

**Chapter 3.3**

P.9 L. 23 To be exact, QCL fluxes are not explained by GC fluxes. They are correlated with GC fluxes.

P.10 L4 What does mean "…no dependency on flux value was observed…" Why should SE depend on flux value? Again, how was SE defined?

P.10 L 14 indicates → indicating

P.10 L 15 forward: "…GC is still useful method to determine soil-atmosphere exchange… at longer time scales.." What is your argument based on? There are no budget calculations in the paper. Averages were reported to be similar, particularly for the small flux regime, but at the same time the fluxes were hardly detected with the GC. Is it correct to say that GC fits for budget studies? How big errors are acceptable in budget studies?

**FIG 2**   add A) and B) to panels and refer to them in the legend

**FIG 4**

- Refer to "A, B, C and D" before each legend text parts; "Figure 4. a) Comparison of N2O fluxes… b) Linear regression…"
- The legend text should be shortened. Remove phrases such as "Also shown is…" Figure 4b is showing 60-min fluxes plotted against 3 min fluxes.
- Please remove the text "Riso campaign 2013, Willow…" from the top of each separate panel and add that part of information into the legend text which is not already there.
- Panel B: indicate what are the two lines in the figure? Why are they not direct lines, but show some tiny variation?
- In Fig. 4 and Fig 5, what is the reason to compare 3-min linear and 60-min exponential fluxes? Why not to compare separately the lin vs exp AND 3-min vs 60 min closure times?

**FIG 7**

- Please use A-D notations, not left/right/upper/lower explanations
- Upper panel: define the lines

**FIG 8**

- An interesting Figure. What does the error bar denote? Is the diurnal variation significant? Why is the hourly data not shown? Are the points averages from many hours? What was actually the frequency of measurements in both campaigns, I did not find it, but I assumed you measured hourly?

---

## Author Response (AR1)

**Response to interactive comment (Anonymous Referee #1) on 'Gas chromatography vs. quantum cascade laser-based N$_2$O flux measurements using a novel chamber design'**

*[R#1.1] Brümmer et al. present a study analyzing linear and non-linear flux calculation methods under high and low flux rates of nitrous oxide and different scenarios of closure time. They use both traditional gas chromatography (GC) (low sampling number during closure time) and high-resolution quantum cascade laser (QCL) sampling. They find that non linear concentration changes are more clearly detectable during high emission scenarios and long chamber closure. Shortening of closure time results in a reasonable agreement between linear (3min) and non-linear (60min) flux estimates, but can only be applied when using the QCL set up. While under low flux conditions, GC measurements result in more scattered flux estimates, in both campaigns mean flux estimates of GC and QCL agreed well. Rare negative fluxes detected by GC measurements seem to be arbitrary and not caused by actual N2O uptake. The paper is well written and a good fit for the journal.*

*However, I could not help thinking that most of the results were as to be expected from literature and not 'radically' new.*

[AC#1.1] We highly appreciate the comments and suggestions given by Anonymous Referee #1. We agree that some results like higher non-linearity in concentration changes under higher emission regimes and longer closure times have been hypothesized and reported earlier. The basic idea of this study is to give a concise overview by showing a side-by-side comparison of QCL vs. GC characteristics, low and high exchange regimes, linear and non-linear flux calculation methods alongside a presentation of our custom-built chamber design. Many other papers, however, usually deal with only a few of the above mentioned components, i.e. either low vs. high fluxes, or only with a GC vs. QCL comparison, or purely with different calculation methods. Therefore, we aimed at integrating the characterization of the measurement system, the exchange regime, and the flux calculation by means of two short campaigns without going into too extensive analyses.

Changes to the manuscript:
None specifically for this comment, but responses to comments R#2.4 and R#2.12 from Reviewer #2 deal with similar topics. See AC#2.4 and AC#2.12.

*[R#1.2] Ultimately, the high temporal resolution of measurements possible with QCL (which do provide more sophisticated ways of flux data processing) and the fact that the concentration measurements are instantaneous, make these measurements desirable for exactly the long-term applications, the authors are suggesting.*

[AC#1.2] The novel QCL application combines multiple advantages over traditional manual sampling systems. These are (amongst others)
- higher temporal resolution of concentration data leading to a higher number of flux rates per day,
- the possibility for robust application of flux calculation procedures,
- easy determination of system malfunction, e.g. caused by insufficiently closed chambers,
- low maintenance for laser operation,

- low uncertainty in flux estimates providing the opportunity for ecological process studies and calculating robust trace gas budgets

For those cases where QCL methodology cannot be applied, e.g. due to high initial investment costs, GC-based measurements may still be useful when investigating longer periods when the focus is not on short-term variability of gas exchange dynamics.

Changes to the manuscript:
None.

*[R#1.3] It is not clear to me, what the accessibility of the described instrumentation is. Is there a plan to make it available for other users, i.e. to 'rent' it out or to make it available within the ICOS project? If that is the case, it should be pointed out more clearly.*

[AC#1.3] We thank the reviewer for this comment. It is a good idea to promote the presented chamber design more clearly as it meets the anticipated standards listed in the ICOS protocol for chamber measurements. That protocol, which will be made publicly available soon by the Ecosystem Thematic Center of ICOS, does not explicitly state precise mandatory dimensions for chamber volume and design, but rather provides size ranges depending on ecosystem type. Information about our chamber system including the construction plan is open to the scientific community and can be requested from the authors. We add the respective information at the end of Chapter 2.1.

Changes to the manuscript:
Sentence added at the end of Chapter 2.1: 'Information about our chamber system including the construction plan is open to the scientific community and can be requested from the authors.'

*[R#1.4] Overall, the most interesting aspect to me is the possibility to study ecological processes in a new way, as shown for the possible net N2O uptake and diurnal variability in emission rates. Interestingly, the study they compare their results to (Shurpali et al. 2016) is mostly an eddy covariance study. It would be interesting if the authors could comment on possible advantages of this automatic chamber against eddy covariance and whether other gases can be sampled in parallel to N2O (I am thinking mostly of CO2, considering the possible coupling of plant activity and N2O emission rates).*

[AC#1.4] One advantage of chamber measurements in comparison with an eddy-covariance approach is the possibility to study small-scale spatial variability of greenhouse gas exchange. This can either be done in natural homogeneous environments or in specific trials at plot scale, e.g. when different types and amount of fertilizers are applied on relatively small plots of a few square meters where the eddy-covariance approach would fail as it requires a homogeneous fetch of up to a few hectares around the tower. Secondly, continuous automated measurements using QCL spectrometry for trace gas analysis like in our study do provide robust estimates of exchange fluxes in situations where assumptions of the eddy-covariance theory are violated. These situations are for example low atmospheric turbulence conditions that frequently occur during nighttime or when measurements are conducted in hilly terrain and advective flows cause significant bias in EC-based fluxes.

As many laser spectrometers that are currently available on the market allow for parallel detection of selected other trace gases – usually $CH_4$ and $CO_2$ – in one analyzer cell, our sampling setup can simultaneously provide concentrations and flux estimates of the chosen greenhouse gases to study coupled environmental processes such as effects of water table, soil moisture and temperature on the respective gases of interest.

We will add the information that parallel detection of different trace gases is possible with most common analyzers in combination with our chamber system.

Changes to the manuscript:

Sentence modified at the end of Chapter 3.4: 'Our study highlights that through its high time resolution QCL-based measurements will not only help enhance process understanding of $N_2O$ exchange by disentangling the strength of different drivers of $N_2O$ production like temperature, soil moisture, nitrogen availability, and microbial activity, but has also the potential to provide new insight into bidirectional exchange characteristics of other trace gases such as $CH_4$, which can be sampled simultaneously with our chamber system depending on analyzer type used.'

**Response to interactive comment (Anonymous Referee #2) on 'Gas chromatography vs. quantum cascade laser-based N$_2$O flux measurements using a novel chamber design'**

*[R#2.1] The authors have tested the performance of a QCL analyzer connected to a new automated chamber, against a "conventional" GC + automated gas sampling unit system. Data from QCL system were used to observe the non-linearity in the concentration increase during the chamber closure. Based on two short campaigns, the paper gives recommendations how should the measurements, data screening and flux calculations be done. The new chamber design is interesting and the system coupled to QCL seems to be fluently producing nice data. Papers presenting new chamber designs, are always welcome, particularly if they can provide generalizations and recommendations which are useful for other chamber operators. The paper is fluently written, and the observation of different patterns in diurnal cycle is interesting and important. However, there are several deficiencies and pitfalls in the data treatment and the argumentation which need revision. The presentation quality would benefit from separating the results and discussion.*

[AC#2.1] We sincerely thank Referee #2 for his/her thorough review. Through the consideration and inclusion of his/her meaningful comments and suggestions, we feel that the manuscript's quality has improved, particularly by shaping the main conclusions and take home messages. We also have streamlined the presentation of the main findings. Below we give our responses to all points raised by the reviewer plus short statements of the actual changes in the manuscript.

Changes to the manuscript:
Main changes are (see specific points below for details):

- Reformulation of the aims of the paper at the end of the Introduction
- Inclusion of a table summarizing main features of the chamber system by providing quantitative measures
- Streamlining the text through splitting up Results and Discussion sections
- Providing a clear story line of investigations (as can be seen by the reformulation of the aims and the newly structured table of contents)
- Shaping up some of the conclusions (see specific points below)
- Few changes to figures (see specific points below)

The restructured aims of the paper now read as follows:
      (1) Presentation of a novel chamber design that is connected to both a vial air-sampling setup with subsequent GC analysis and a QCL spectrometer
           ■ Description of design and setup in Sections 2.1 and 2.2
           ■ New chamber system is used for the following investigations (aims 2 to 5)
      (2) Characterization of the shape of the concentration increase
           ■ → Is the shape rather linear or non-linear?
           ■ Quantification of the curvature ($\kappa$) in concentration increase
           ■ Using $\kappa$ to verify chamber density
               ● → Is $\kappa$ dependent on wind speed, wind direction, on the flux itself or on closure time?
      (3) Comparison of N$_2$O fluxes and their associated standard errors from linear and non-linear regression models
      (4) Testing the novel chamber system under high and low flux conditions and comparing GC vs. QCL-based flux estimates
      (5) Investigation of ecosystem and climate-specific flux characteristics such as N$_2$O uptake and diurnal variation

The single paragraphs of the Results and Discussion section are now as follows:

3. Results
    3.1    Shape of concentration increase and curvature ($\kappa$) determination
    3.2    Comparison of $N_2O$ fluxes and their associated errors from linear and non-linear regression models
    3.3    GC vs. QCL-based fluxes under high and low exchange regimes
    3.4    $N_2O$ uptake and diurnal variation
4. Discussion
    4.1    The parameter $\kappa$ as a chamber performance criteria
    4.2    Closure time and measurement frequency – How long and how often is enough?
    4.3    Differences between GC and QCL-based fluxes
    4.4    Enabling investigations of flux pattern characteristics

*[R#2.2] First, the performance of the GC sampling system makes me wonder whether the comparison of two systems is meaningful. Before making any comparisons, the authors should find out the reason for the bad performance of the GC.*

[AC#2.2] Please see detailed response to [R#2.6].

Changes to the manuscript:
Please see detailed response to [R#2.6].

*[R#2.3] Secondly, there are several conclusions in the paper which are just qualitative, and as such they are vague and are not supported by the presented data.*

[AC#2.3] We fully agree that some conclusions came a bit out of the blue. We have modified the respective sections as outlined in the specific comments to [R#2.7], [R#2.9], [R#2.11], [R#2.33], [R#2.34], and [R#2.39] below.

Changes to the manuscript:
See responses to comments [R#2.7], [R#2.9], [R#2.11], [R#2.33], [R#2.34], and [R#2.39] below.

*[R#2.4] Third, I share the worry of the first reviewer that most of the results shown here are already well known. For example, it has been reported already in numerous papers that the curvature in concentration increase is higher with longer closure time, and that using the linear calculation instead of non-linear can result in great underestimate in flux rate. Instead of reporting curvature, it would be more useful to quantify what is the limit of curvature after which the authors recommend the use of non-linear fitting method.*

[AC#2.4] We agree that higher curvature in concentration increase at longer closure time has been hypothesized and shown before. But to our knowledge, it hasn't been quantified and neither its dependency on $N_2O$ fluxes (Fig. 3A) nor on chamber performance criteria like the insensitivity towards wind speed and direction (Fig. 3C and 3D) has been explicitly analyzed like in our study. This is clearly a new investigation alongside presenting a novel chamber design. Also, flux underestimation when using linear instead of non-linear regression may certainly be true for GC measurements when only a limited number of samples are available. But the point in our paper is that we on the hand highlight the advantages of QCL measurements (high time resolution, low standard errors of fluxes; *cf.* Figs. 2, 4, 5, 6) and then recommend to reduce chamber closure time to be able to apply linear regression (see modified Fig. 4D for better visualization of low differences between the application of linear

vs. non-linear regression for flux calculation). We also attach here graphs showing the flux difference, i.e. non-linear−linear, plotted against curvature (Fig. R1). Note that during shorter closure time (10 min; blue circles), relatively small (absolute) differences between the two calculation methods occur, although curvature was highly variable and single $\kappa$ values up to −1000 $\mu$g N m$^{-3}$ h$^{-2}$ were found.

We fully agree that the manuscript would benefit from streamlining the aims, results, and messages towards a more concise overview of useful take home conclusions for the reader. See 'Changes to the manuscripts' at [AC#2.1] for the main modifications. However, regarding curvature in this study, it should not be used to define a threshold after which linear over non-linear flux calculation should be used as it is supposed to demonstrate chamber performance criteria as highlighted in Fig. 3C and 3D. Together with the new Fig. 4D and its slope close 1 in the most common flux range between 0 and 200 $\mu$g N m$^{-2}$ h$^{-1}$, we prefer using the argument throughout the manuscript that reducing chamber closure time and applying linear regression for flux calculation is a valid approach.

[Figure]

New Figure 4D: Linear regression analysis of $N_2O$ fluxes <200 $\mu$g N m$^{-2}$ h$^{-1}$ with adapted regression from the exponential vs. the linear model.

[Figure]

Figure R1 (will not be shown in the manuscript): Panels A, B, and C: Dependency of $\Delta F$, i.e. $N_2O$ fluxes from non-linear regression–linear regression, on $\kappa$ values for different ranges. Panel D: Dependency of normalized flux difference on $\kappa$ values.

Changes to the manuscript:
See [AC#2.1] plus revised Fig. 4D and 5.

*[R#2.5] Also, it would have been interesting to learn more about the advantages and possible problems in the "novel" chamber design. In general, the paper could be more valuable would it provide more quantitative information and recommend some general tests which each chamber operator should run to ensure adequate data quality. It is also a bit questionable if the paper with such a short piece of data (25 + 6 days) is enough the draw firm conclusions. See more comments below.*

[AC#2.5] We highly appreciate this comment and agree that a concise overview of system features will help the reader to get familiar with chamber and instrumentation characteristics. These information including flux detection limit, closure time, number of daily cycles, sampling frequency, etc., are summarized in Table 2. Other more qualitative features are given in Section 2.1. A general test every operator should perform is a (somewhat indirect) density test with a calculation of standard errors of fluxes under different flux magnitudes where the shape of the concentration increase/decrease appears to be valid such as in Fig. 2B at DOY 107.8. The standard errors of these 'good fluxes' should be taken as a reference. Operators should inspect all fluxes that deviate largely from those reference values. However, these absolute numbers predominantly depend on the precision of the analyzers that are used, thus making it difficult to provide specific thresholds of errors after which flux values should generally be discarded. In our study, QCL-based fluxes with a standard error >3 µg N m$^{-2}$ h$^{-1}$ have undergone further double-checking. Only a few of those remained

plausible as can be seen in Fig. 4C. Regarding the 'short piece of data', we think that the value of a methodological study is not necessarily depending on the length of the observation. In fact moving the same systems to different places instead of measuring longer at one site, increased the range of test conditions, which has added value to the study and made its conclusions more robust. The conditions varied from very low to high fluxes, high external wind speeds (Risø), moderate wind speeds (Braunschweig), lower and higher temperatures. The effects were clear and thus we feel that performing longer tests would not have considerably increased the information with respect to the objectives of the study.

Changes to the manuscript:
Inclusion of Table 2 with quantitative features; we also add some rather qualitative characteristics in Section 2.1 such as the fact that the size of the chamber allows for investigations including plants of considerable size (even up to rape seed; publication in preparation) and that lifting the chamber diagonally away from the soil frame reduces shading for radiation and precipitation, thereby keeping the measurement spots as natural as possible.

Table 2: Features of the chamber-analyzer system used in this study.

| | GC[*]
(model: Shimadzu GC-2014) | QCL[*]
(model: Aerodyne Research Inc. mini-QCLAS) |
|---|---|---|
| No. of chambers | 3 | 3 |
| Chamber closure time | 60 min | 60 min
10 min (recommended) |
| Sampling frequency | every 20 min | 0.1 sec (max)
5 sec (recommended) |
| No. of concentration records per chamber run | 4 | 36000 in 60 min
6000 in 10 min |
| No. of chamber cycles per day | 24 (max) | 72 (recommended)
144 (max) |
| Maximum number of samples | 168 (depending on autosampler size) | Limited only by data storage capacity of QCL's computer or external hard drive |
| Lag time | (~10 sec) | ~10 sec |
| $N_2O$ flux detection limit ($\mu g\ N\ m^{-2}\ h^{-1}$) | 13.0 | 2.6 |
| Mean campaign $N_2O$ flux ($\mu g\ N\ m^{-2}\ h^{-1}$) | BS (pref.[1]): 6.42
Risø (pref.[1]): 77.40 | BS (lin.): 7.77
Risø (lin.[2]): 122.95 |
| Mean campaign SE of $N_2O$ fluxes ($\mu g\ N\ m^{-2}\ h^{-1}$) | BS (pref.[1]): 5.98
Risø (pref.[1]): 8.17 | BS (lin.): 0.13
Risø (lin.[2]): 0.21 |
| Median campaign $N_2O$ flux ($\mu g\ N\ m^{-2}\ h^{-1}$) | BS (pref.[1]): 5.15
Risø (pref.[1]): 64.80 | BS (lin.): 7.38
Risø (lin.[2]): 105.43 |
| Median campaign SE of $N_2O$ fluxes ($\mu g\ N\ m^{-2}\ h^{-1}$) | BS (pref.[1]): 5.04
Risø (pref.[1]): 4.72 | BS (lin.): 0.10
Risø (lin.[2]): 0.17 |
| Percentage of flux estimates where HMR could be fitted | BS: 8.5 %
Risø: 37.9 % | BS: 100 %
Risø: 100 % |

*GC – Gas chromatograph, QCL – Quantum cascade laser spectrometer, [1]preferred means non-linear HMR model was used if applicable, otherwise robust linear regression was taken, [2]mean/median of DOY 105.5 to 108.5 to make it comparable to GC data set*

MORE DETAILED COMMENTS:

*[R#2.6] What is the reason for the very bad performance of the GC system? On p5 lines 7-10 it is said that the system was checked against ten samples of ambient air, and only if the CV falls <3%, the data is acceptable. Is this CV limit of 3% really acceptable for a GC system? From Figs. 2a and 7a it seems clear that the GC is not able to resolve concentration increases for fluxes < 20 µgN m-2 h-1. At least to my knowledge, much lower fluxes analyzed with the GC have been reliably reported. I think that a comparison between QCL and GC is not really meaningful if GC is not able to measure these "small" fluxes of N2O. However, Fig. 2a makes me doubt, whether the problem is in the autosampler, and not in the GC detection limit? In some cases the GC can quite perfectly detect a concentration increase of about 12 ppb's similarly to the QCL (second measurement of DOY 339), but during many other closures the data seems arbitrary. What is the reason for that?*

[AC#2.6] We appreciate the reviewer's concerns, but do not agree with the premise that a CV below 3 % (it was mainly close to 2 % in our study) at ambient concentrations is a 'very bad performance'. Inter-laboratory comparisons within Germany have shown that GC systems commonly exhibit CVs in this range during routine operations (publication in preparation). Also compare with Parkin et al. (2012), who show a CV of 4.4 % as an example in their Fig. 2. Based on this they calculated a detection limit of about 35 ppb $h^{-1}$ for the linear flux model (see their Fig. 6, corresponds to about 40 µg N $h^{-1}$) and even higher detection limits for non-linear flux calculation schemes (which however reduce bias). Furthermore, detection limits should be determined based on statistics and not based on single flux measurements (e.g., on DOY 339). The median standard error of GC based flux measurements in our campaigns was SE = 6.5 µg N $m^{-2}$ $h^{-1}$, thus the detection limit is approximately DL = 2 * SE = 13.0 µg N $m^{-2}$ $h^{-1}$. See also response to Short Comment 3 for details.

Changes to the manuscript:
None.

*[R#2.7] There are conclusions in the paper which are not supported by the presented data. For example: p.1 line 30: "new chamber design reduces the disturbance of the soil". There was nothing on that in the results. What are the possible disturbances? How can you detect that those can be omitted by your system?*

[AC#2.7] This statement is simply related to the way the chamber is lifted and dragged away from the collar spot in a 45° angle. In comparison to many other chamber designs, soil and vegetation inside the soil collar are thereby kept under as natural conditions as possible, because the positions of the chambers when they are not operating largely prevent unintended shading and do not disturb throughfall, which is important when the chamber system is supposed to run for a longer time. This information was already given at the end of Section 2.1. We will slightly rephrase the sentence in the Abstract.

Changes to the manuscript:
Sentence modified to: 'Our new chamber design prevents the measurement spot from unintended shading and minimizes disturbance of throughfall, thereby complying with high quality requirements of long-term observation studies and research infrastructures.'

*[R#2.8] Or: lines 25-26: GC was found to be a useful method to determine N2O fluxes at longer time scale". Where is the data to prove such a conclusion? There were no budgets*

*calculated. What happens with the low fluxes, how can you reliably determine budget if you cannot detect the flux?*

[AC#2.8] See response to [AC#2.39].

Changes to the manuscript:
See response to [AC#2.39].

*[R#2.9] Or p.8 line 31 forward: how do you justify the recommendation of removing the first 2 minutes of data?*

[AC#2.9] We agree that the reader must have been puzzled by this sudden recommendation without showing any data. This statement arises from an observation we made in the increase pattern of the concentrations. In ~5 % of the cases, a somewhat irregular pattern as shown in the figure below was observed. It only happened right after setting the chamber onto the soil collar so maybe it was caused pressure fluctuations. We could not identify any correlations to either environmental or internal system conditions when this pattern was found. We therefore think it is a reasonable security procedure to remove the first two minutes (because it never exceeded this initial period) of data from a chamber cycle to ensure natural steady state soil efflux.

[Figure]

Fig. S1: Example of $N_2O$ concentrations right after chamber closure up to 0.1 h (=6 minutes). Note the small dent at the beginning up to 0.03 h (=108 seconds).

Changes to the manuscript:
Inclusion of Fig. S1 into the supplementary material and reference to the figure on Page 9, Line 1.

*[R#2.10] Or "1 to 5 s frequency was sufficient to keep SE on much lower level than in fluxes determined by the GC method" (p. 9). "sufficient" was not defined here.*

[AC#2.10] See detailed responses to [AC#2.33] and [AC#2.34].

Changes to the manuscript:
See modified Fig. 5 detailed responses to [AC#2.33] and [AC#2.34].

*[R#2.11] How do you justify the limit of using only the first 10 min of the data? Please give some argument based on the data, not just the feeling that this is good.*

[AC#2.11] In [AC#2.9] we now point out that the first two minutes of data after chamber closure should be discarded and not used for the regressions. One of the main conclusions of the paper is that applying linear regression to only a short piece of QCL data is fully sufficient to reliably calculate the flux. We show this for periods of three minutes. Chamber operators can decide on their own whether they want to use 3 or 5 or 10 minutes for flux calculation or even extend the initial data that is discarded. Our point is that we clearly found that it does not take a long chamber deployment time to calculate robust fluxes. A period of 10 minutes gives the user enough tolerance for setting its own schedule. We will clarify this in the respective paragraphs of the manuscript.

Changes to the manuscript:
See [AC#2.9] and analyses of fluxes from 3-min linear regressions (Figs. 4, 5, 6, 7). Clarification will be provided in the newly arranged Section 4.2.

*[R#2.12] Many of the conclusions of the paper follow those observed in previous studies and are already well known. The one exception is the data in Fig. 8 showing the diurnal variation in N2O flux, as this phenomenon is not much studied. In addition to such data providing information on GHG formation processes, the value of the paper could have been in showing how exactly this chamber system works and what are the special and quantified conditions needed to run the system and to screen the data in order to provide reliable flux data.*

[AC#2.12] See responses to [RC#2.4] and [RC#2.5].

Changes to the manuscript:
See responses to [RC#2.4] and [RC#2.5].

*[R#2.13] I strongly recommend to separate the results and discussion parts; presently it is difficult to follow the storyline.*

[AC#2.13] We agree that splitting results and discussion into two parts may improve the readability of the paper.

Changes to the manuscript:
See [AC#2.1] for the newly structured results and discussion sections.

METHODS

*[R#2.14] P.4 L.3 Why "semi-automatic"?*

[AC#2.14] The term 'semi-automatic' refers to the operation mode when the system is connected to vial air-sampling, i.e. collecting air in the autosampler. It describes the fact that the sampling is automatic, but the actual gas analysis is done later in the lab. The term is explained in the Introduction, P.2, L.24.

Changes to the manuscript:
None.

*[R#2.15] P.4 L.6-7 A volume of L x W x H does not result in 0.33 m3*

[AC#2.15] It is true that the interior dimension we have given on Page 4, Line 6 would result in ~0.341 $m^{-3}$; however, the number 0.33 $m^{-3}$ describes the real conditions as we needed to

subtract volume of some items inside the chamber such as the fan, different bigger screws and supporting racks and tubes. We clarify this in the manuscript.

Changes to the manuscript:
Sentence modified to: 'Subtracting inside items such as an axial fan, screws, supporting racks and tubes, the chambers have a headspace volume of 0.33 $m^{-3}$ and covered a surface area of 0.56 $m^{-2}$.'

Chapter 2.3:

*[R#2.16] p.5 L. 26: what are the conditions when HMR function cannot be fitted? ; L27, what is Akaike information criterion, please open this a bit, although there is the reference, the reader should get some kind of an idea just by reading the text here.*

[AC#2.16] For clarification, we slightly rephrase this paragraph and provide some additional information as given below.

Changes to the manuscript:
The paragraph on Page 5, Lines 24 ff. has been modified to: 'Briefly, non-linear flux estimation with the HMR method (R Core Team, 2012; HMR package version 0.3.1) was performed when four data points were available and all of the following criteria were met, i.e. (1) the HMR function could be fitted, (2) Akaike information criterion (AIC; Burnham and Anderson, 2004), which is a measure of (relative) model quality, i.e., gives fit quality penalized by the model's degrees of freedom, and can be used to compare the quality of different model fits to the same dataset, was lower for HMR fit than for linear fit, (3) $p$ value of flux calculated using HMR was lower than that from robust linear fit, and (4) the HMR flux was less than four times larger than the robust linear flux. Otherwise, robust linear regression or ordinary linear regression was used when four or three data points were available, respectively.'

*[R#2.17] Equations 1-3 and the text related to them: add units.*

[AC#2.17] To keep fluent readability, we will add units in a single sentence at the end of Section 2.3.

Changes to the manuscript:
Sentence added: 'Units for concentrations $c(t)$, $c_{max}$, and $c_0$ are g $m^{-3}$, units for $k$ are g $m^{-2}$ $s^{-1}$, and units for $\kappa$ are g $m^{-3}$ $s^{-2}$.'

RESULTS

*[R#2.18] P.7 L.7: "low negative k values": care should be taken to express the relations between negative and more negative values. Perhaps more clear to speak about absolute values when comparing these.*

[AC#2.18] We agree that this expression may lead to confusion and will refer to absolute values as suggested.

Changes to the manuscript:
Sentence changed to: 'Extremely low absolute $\kappa$ values between $-10^{-4}$ and $-10^0$ – indicating quasi-linearity in $\partial c/\partial t$ – were almost exclusively found under low flux conditions, whereas…'

*[R#2.19] P.7. L 10-11 "Near zero fluxes indicate no considerable changes in N2O concentration". Isn't this self-evident without any measurements? Also, what is "considerable change in N2O concentration"? Do you mean significant? If there's no significant increase in concentration, there is no flux, true? Remove or reword the sentence. The whole chapter (Lines 7-14) seem quite self-evident, as the authors hint in the last sentence of the chapter. From Line 15 onwards you say that application of linear model is acceptable in some cases. However, no quantification, i.e. limit below which this is acceptable, is given. I also do not understand how do you draw this conclusion from the results on Lines 7-14.*

[AC#2.19] We agree that it would be self-evident when only taking the cited part as given above. In the manuscript, however, the sentence is clearly written in the context of the kappa discussion. Also, if there is no significant (with regard to being lower than the flux detection limit) increase in concentration, i.e. the flux is (close to) zero, which depicts a very important state of the ecosystem and should definitely be taken into account, the corresponding curvature is also marginal. In our opinion, this should at least be stated once. Again, a quantification of kappa hasn't been shown many times before and is used in our paper as a chamber performance criterion (*cf.* Fig. 3C and 3D; [AC#2.4]). We definitely like to stick with the description in Lines 7-14 as it is. Regarding the statement about the acceptance of linear regression at low fluxes, we will include the newly found relationship of Fig. 4D and rephrase the statement accordingly (see below).

Changes to the manuscript:
Sentence starting on Page 7, Line 15 modified to: 'Our results imply that at low to moderately high flux rates <200 µg N m$^{-2}$ h$^{-1}$ (*cf.* Fig. 4D) and/or short chamber closure, the slight non-linearity in concentration change when calculating fluxes is of minor importance and the application of linear models is acceptable, particularly with regard to other commonly observed errors such as those originating from soil disturbance, chamber placement (Christiansen et al., 2011), temperature, pressure and humidity perturbations, etc. (Parkin and Venterea, 2010).'

*[R#2.20] P.7. L.23: what is meant with dispersion here?*

[AC#2.20] The soil surface basically releases a dispersion plume to the chamber headspace, which eventually is being transported through tubing to the analyzer. If the dispersion of the elevated gas concentration is initially not uniformly mixed with the air inside the tubing, then a lagged concentration increase in the form of exponential analyzer readings (up to a certain point in time) may be observed.

Changes to the manuscript:
Additional information to the sentence starting on Page 7, Line 22: 'These are exponentially increasing N$_2$O concentrations after chamber closure due to possible dispersion effects leading to biased analyzer readings when the elevated gas concentration is initially not uniformly mixed with the air inside the tubing, placement of…'

*[R#2.21] P.7 L. 29-30 "…outside the chamber and inside chamber conditions…" please reword*

[AC#2.21] Sentence rephrased.

Changes to the manuscript:
Changed to: 'We also investigated the possible effect of ambient wind speed and direction on concentration build up characteristics (Figure 3C and 3D, respectively) as differences between the turbulence conditions outside the chamber may possibly vary from those conditions inside the chamber under changing wind speed.'

*[R#2.22] P.7 L. 30-31 "…coupling of the flux under ambient conditions…" I do not understand this sentence, please reword*

[AC#2.22] It means that placing a chamber on soil is a substantial interference with the local wind regime, particularly when wind speed is high and soil pores in the uppermost soil layer may have been ventilated under ambient conditions (i.e. conditions without a chamber) and it thus would take a while until a steady state flux is established. Sentence rephrased.

Changes to the manuscript:
Sentence(s) changed to: 'Theoretically, pores in the uppermost soil layer might be ventilated under high wind speed when no chamber is in place, thus a close coupling of the flux to the atmosphere exists. Consequently, the establishment of a steady state flux may be more postponed under these high wind speed conditions once the chamber is put onto the soil frame.'

*[R#2.23] P.7 L.28 → What about the impact of the fan speed on curvature? Soil pores may be ventilated also by the fan (see for example Lai et al. 2012, BG).*

[AC#2.23] Lai et al. (2012) found that 13 min of closure were needed before their fluxes (concentration increase) became constant and therefore they extended the deployment period to 30 min. In our study, we found in most cases clear linear or slightly saturating concentration increases right from the beginning. The few cases with 'irregular start patterns' are discussed under [AC#2.9] and in Section 4.1.
Not only fan speed, but also orientation may affect natural efflux from soil. Information on our fan operation is added to Section 2.1.

Changes to the manuscript:
Sentence on Page 4, Line 13 is modified to: 'Chambers were ventilated during measurements using an axial fan, which was mounted to produce a horizontally oriented airflow alongside chamber walls to minimize interference with the natural steady-state soil efflux, but to maximize proper mixing of the chamber headspace as was described in Drösler (2005).' See also [AC#2.9] with regard to the removal of the initial 2-min period.

Chapter 3.2

*[R#2.24] Might be good to start with your own results, not with the literature review. For clarity, I strongly recommend separating results and discussion.*

[AC#2.24] Results and discussion will be split up.

Changes to the manuscript:
See [AC#2.1] for newly arranged Results and Discussion section.

*[R#2.25] P.8 L. 15 onwards: Figure 4B indicates that actually the 3-min/lin method produces higher fluxes than the 60 min/exp method in the lower flux regime (below 200 µg N m-2h-1). When taking into account also the higher fluxes (n=6), the relationship changes so that these 6 data points make a very strong impact, as the authors already discuss. Even though this is the case, the discussion here emphasizes continuously how the linear fluxes are smaller than exponential, although the results shown support this observation only for the few high flux points. This makes me to doubt how one can make generalizations about the validity of these two methods. I am missing discussion which tries to find explanation for the higher*

*fluxes with 3-min/lin method. Is it so that the data set should be split, or is it far too small to make generalizations?*

[AC#2.25] Looking at the newly arranged Fig. 4D (see under [AC#2.4] and [AC#2.44]), there is no indication that 3-min-lin fluxes result in systematically higher values than 60-min-exp fluxes. Only few data points are under the 1:1 line and the slope is 0.989. It is true that we discuss flux underestimation by using linear methods, but that is exactly what is mainly found in the literature (and caused the regression in Fig. 4B). We do not follow the quest of trying to explain why the 3-min-lin method produces higher numbers than the HMR method, because it is just simply not the case, neither in literature nor in the data shown in Fig. 4. We will, however, give a more detailed discussion why HMR-based fluxes are sometimes higher under high flux conditions.

Changes to the manuscript:
Discussion added in Section 4.2. Beside any form of unintended interferences to the 'natural steady-state flux' like for example disturbances through macrofauna, fluctuating pump performance or analyzer malfunctions due to internal re-calibration during chamber deployment, much higher 60-min-based HMR fluxes compared to 3-min-based linear fluxes may be observed when one of two following concentration increase patterns are observed.

1) Slow initial increase of concentrations followed by steeper rise after some minutes. Slope of the linear fit will then be much lower than the one from the HMR fit (lin fit at $t_0$).
2) Steady linear start of concentration increase followed by sudden relatively sharp bend with lower linear increase afterwards. HMR fit will also have a much steeper slope at $t_0$ than the linear fit, which will be on top of the data points for the first few minutes.

Red dots in Fig. 4B indicate situations similar to those described under (2) above.

[Figure]

Panel B of Figure 4: Linear regression analysis of $N_2O$ fluxes from the exponential vs. the linear model. Red dots in Fig. 4B indicate situations where a steady linear start of concentration increase was followed by a sudden relatively sharp bend with lower linear increase afterwards.

*[R#2.26] Would be also interesting to see, what happens to the SE/RMSE or similar, when apparently low fluxes are calculated with the exponential method. Is there perhaps a risk of*

*higher error /noisy flux data? Would it be possible to find a flux rate below which the linear method is working more reliably than the exponential?*

[AC#2.26] See discussion above. For fluxes <200 µg N m$^{-2}$ h$^{-1}$ there is no significant deviation between the two methods. Even for the few high flux rates, standard errors are still on an acceptable level around 3 µg N m$^{-2}$ h$^{-1}$.

Changes to the manuscript:
None in particular for this comment, but see also responses [AC#2.4] and [AC#2.25].

*[R#2.27] L. 25 I do not understand this sentence*

[AC#2.27] What sentence and what is unclear? In the study by Kroon et al. (2008), linear flux rates were underestimated by 60 % compared to those from an exponential function. This was the same order as the flux uncertainty due to temporal variation. Or was the following sentence meant? A simple description of mean and median values of standard errors of fluxes from both the linear and the non-linear model is given.

Changes to the manuscript:
As it is unclear to us what is meant, we stick with the given formulation as we feel the description is very clear.

*[R#2.28] P.8 L.31 onwards, continuing on P.9: Here you give important recommendations, but show no data. Also the reference to Section 3.1 is strange, as I do not find anything about the delayed concentration increase in 3.1. This data should be definitely shown if such recommendations are given. You should justify the removal of the first 2 minutes of data: why exactly 2 minutes?*

[AC#2.28] See [AC#2.9] for the reason of the removal of the initial 2-min period. The reference to Section 3.1 is indeed strange, because it is a relic from a former version when the kappa analysis looked slightly different than in the submitted version and will be removed.

Changes to the manuscript:
Inclusion of a graph showing delayed concentration increase for the justification to remove the first two minutes of data. Reference to Section 3.1 removed.

*[R#2.29] P.8 L3. "…we also compared HMR-based fluxes from QCL? with robust linearly calculated…". How does this vary from that in Fig 7 upper right panel?*

[AC#2.29] We assume the reviewer refers to Page 9, Line 3. The difference between the comparison described in Lines 3-10 and Fig. 7 (upper right panel) is that in Fig. 7 linear fluxes are based on 3 minutes of data (as everywhere else in the manuscript), whereas in the described comparison in Lines 3-10 (data not shown), the linear fluxes are – as mentioned in Line 4 – based on the full 60-min cycle of data. To avoid misunderstanding, we add information to the caption of Fig. 7. Please note that in the entire manuscript linear fluxes always refer to 3 minutes of data and HMR fluxes always refer to the full available data set (60 min in Braunschweig and in Risø from DOY 105.5 to DOY 108.5 and 10 min in Risø before DOY 105.5 and after DOY 108.5). This information is explicitly mentioned on Page 5, Lines 30-33.

Changes to the manuscript:

Modified Fig. 7 caption (other changes to Figure 7 are given in [AC#2.46]: 'Panels A and B: GC vs. QCL-based $N_2O$ fluxes. Panels C and D: Relationships between standard errors (SE) of $N_2O$ fluxes and the respective flux values. Blue markers indicate QCL data, which are all based on the 3-min linear calculation method. Red markers indicate GC data, which are based on the full 60-min data set. Crosses are plotted for GC data when all criteria for flux calculation using the exponential HMR model were met (see text for details), otherwise circles are plotted indicating the usage of a linear model for flux calculation.'

*[R#2.30] A general comment/hint: there are many different comparisons with different analyzers, calculation methods and closure times, in which partly different data sets have been used (low and/or high flux) and it is not easy to follow how do all these small experiments differ from each other or support each other. A separate result section with subsections dedicated to each of these questions might help in that. Now there is lot of text (e.g. Chapter 3.2) and it is difficult to follow the argumentation on logics of the text. Also a clearer division into paragraphs would help the reader. And, as already pointed out, division into results and discussion is needed.*

[AC#2.30] We fully agree and split Section 3 into two parts as outlined under [AC#2.1].

Changes to the manuscript:
Results and discussion will be split up.

*[R#2.31] P.9 L. 3-10: In which figure are these shown? Are the slope of 0.97 (lin fluxes are independent) and the HMR fluxes being 22% higher in conflict with each other? How is it possible that now the linear and HMR based fluxes estimated from 60-min data are almost identical (slope=0.97), while earlier you have stated that linear method underestimates the fluxes?*

[AC#2.31] See [AC#2.29]. The analysis described in Lines 3-10 is not shown in a figure as mentioned in Line 4. The whole idea of this paragraph is to make these values, i.e. linear fluxes from 60-min closure comparable to other results presented in literature as for example Kroon et al. (2008) or Forbrich et al. (2010), which have been included in the discussion on Page 8, Line 19 ff.

Changes to the manuscript:
See [AC#2.29].

*[R#2.32] P.9 L 11-18: How were the standard errors calculated?*

[AC#2.32] In the whole manuscript, we deal with the standard error of the flux, not of individual concentrations. As the flux is a parameter in Equation (1), SE is the standard error of the parameter in the respective regression model (not of the residuals of the concentrations). The regression algorithm used is based on the Levenberg-Marquardt method and was taken from the R package 'minpack.lm' (https://cran.r-project.org/web/packages/minpack.lm/minpack.lm.pdf), function 'nlsLM'. The parameter errors are provided by the algorithm. Further details can be found in:
- https://www.rdocumentation.org/packages/minpack.lm/versions/1.2-0/topics/nlsLM
- Equation 22 in http://people.duke.edu/~hpgavin/ce281/lm.pdf
- Bates, D.M. and Watts, D.G.: Nonlinear Regression Analysis and Its Applications, Wiley, 1988.
- Moré, J.J.: The Levenberg-Marquardt algorithm: implementation and theory, in Lecture Notes in Mathematics 630: Numerical Analysis, G.A. Watson (Ed.), Springer: Berlin, 1978, pp.105-116.

Changes to the manuscript:
Additional information added at the end of Section 2.3: 'Standard errors in this study were calculated as the parameter errors from the respective regression model with the algorithm being based on the Levenberg-Marquardt method ('nlsLM function in R package 'minpack.lm', R Core Team, 2012).'

*[R#2.33] This section and Figure 5: I think that SE is not an appropriate quantity when estimating the "sufficient" frequency of concentration data. By definition, SE is related to (the root square of) the number of observations. It is therefore evident that if you decrease the frequency and the number of data, you increase the SE. In a case where the random error of the concentration measurement during the chamber closure is constant, the SE will anyway increase in case the number of observations decreases, whereas the NRMSE, or the error in the flux will not increase. Therefore a better quantity to estimate the error related to the frequency of concentration data is RMSE (or NRMSE).*

[AC#2.33] We sincerely thank the reviewer for this catch and fully agree that reducing sample size automatically leads to an increase in the error estimate when using the method given in [AC#2.32]. However, instead of taking RMSE – which wouldn't work, because we deal with the SE of a parameter and not of residuals – we normalized the SE by multiplication with $\sqrt{n}$. This is now shown in the newly arranged Fig. 5 (see below). The result is fascinating: mean and median for both 3-min-lin and 60-min-exp fluxes are basically invariant with changing sampling time even up to a frequency of 0.03. Note that only 6 data points are left in that latter frequency class for the 3-min-lin fluxes. We rephrase our conclusions accordingly.

Changes to the manuscript:
Inclusion of newly arranged Fig. 5 and reformulation of the paragraph on Page 9, Lines 11-19: 'A further intriguing analysis shows that standard errors were found to be invariant on QCL sampling frequency (Figure 5). We simulated different sampling times ranging from one tenth of a second to 25.6 sec, which corresponds to a frequency of 0.0390625 Hz, by excluding the respective intervals from the original 10-Hz dataset. Results show that the median of the standard error of the fluxes remains stable over a wide range of measurement frequencies. At a frequency class of 0.15 and lower (3 boxes on the right-hand side of Fig.5), which corresponds to a sampling time of ~5 sec and higher, lower and upper quartile values begin to deviate and the median changes slightly.' (can now be found at the end of Section 3.2). Further, we add at the end of Section 4.2: 'The conclusion we can draw from this finding is that chamber operators – in case an analyzer with a precision like the QCL presented in this study is available – can reduce their sampling time down to 5 seconds without risking an increase of the standard error of the flux, which would still be on a much lower level than those obtained from GC measurements.'

[Figure]

Figure 5. Boxplots of standard errors of $N_2O$ fluxes for different frequency classes and regression models used, i.e. linear regression with 3 minutes of data (upper panel) and the exponential HMR model with 60 minutes of data (lower panel). To avoid a pseudo-dependency on sample size, the standard errors were normalized by multiplication with $\sqrt{n}$. Black squares represent the arithmetic mean, red horizontal lines indicate the median, blue horizontal lines indicate lower and upper quartile values, black whiskers represent the interquartile range and outliers from this range are plotted as grey crosses.

*[R#2.34] Your argument "..sampling times between 1 and 5 sec are sufficient to keep SE of fluxes on a much lower level…" is vague. How do you justify that exactly the 1-5 sec limit is sufficient? What means sufficient? How much is "much lower level"? Please quantify and justify this with an appropriate and objective criteria.*

[AC#2.34] See [AC#2.33]. The threshold will be set to 5 seconds.

Changes to the manuscript:
See [AC#2.33]. Rephrasing of occurrences where the threshold is given; now set to 5 seconds.

*[R#2.35] P.9 L.12: "..to approx.. one minute,…" isn't it approx. half a minute (25.6 sec)?*

[AC#2.35] Correct, but the paragraph has been rephrased anyway as mentioned under [AC#2.33].

Changes to the manuscript:
Rephrased as mentioned under [AC#2.33].

*[R#2.36] P.9 L. 23 To be exact, QCL fluxes are not explained by GC fluxes. They are correlated with GC fluxes.*

[AC#2.36] We agree. The expression would fit better if we would look at a controlling factor (x-axis) of some dependent variable (y-axis). Sentence rephrased.

Changes to the manuscript:
Sentence rephrased to: 'A linear regression revealed no significant relationship between GC and QCL fluxes with a very low coefficient of determination of 0.036 (Figure 7A).' We also rephrased the sentence on Page 9, Lines 31-32 '48 % of the variance in QCL-based fluxes could be explained by fluxes from the GC method.'. It now reads 'A linear regression between GC and QCL fluxes revealed a coefficient of determination of 0.48 (Figure 7B).' as it deals with the same topic as the one above.

*[R#2.37] P.10 L4 What does mean "…no dependency on flux value was observed…" Why should SE depend on flux value? Again, how was SE defined?*

[AC#2.37] The standard error of a flux may depend on the flux itself for example when at very low fluxes (low concentration increases) the slope fit may be prone to much higher uncertainty than at larger fluxes when an analyzer with moderate or low precision is used. On the other hand high fluxes may show high standard errors for example when an analyzer is not well calibrated or not able to properly resolve certain concentration ranges. This is something we needed to investigate and a dependency that might explain faulty QCL calibration could not be found. See [AC#2.32] for SE calculation method.

Changes to the manuscript:
None.

*[R#2.38] P.10 L 14 indicates → indicating*

[AC#2.38] No, we don't change that. The word 'indicates' refers to 'The fact that…'. We think changing this to 'indicating' would lead to incorrect grammar (and/or different meaning).

Changes to the manuscript:
None.

*[R#2.39] P.10 L 15 forward: "…GC is still useful method to determine soil-atmosphere exchange… at longer time scales.." What is your argument based on? There are no budget calculations in the paper. Averages were reported to be similar, particularly for the small flux regime, but at the same time the fluxes were hardly detected with the GC. Is it correct to say that GC fits for budget studies? How big errors are acceptable in budget studies?*

[AC#2.39] We agree that 'at longer time scales' is a bit strong given the fact that we only show a few weeks of data. Nevertheless we need to point out that mean and median of the whole BS campaign where fluxes were on average quite low match pretty well. In Risø – although single flux values were closer to each other – deviation between GC and QCL mainly occurred at high fluxes (Fig. 7B) under the influence of fertilization. But this also

indicates that using a GC is still useful for a wide range of periods over an entire year. However, taking into account that the bulk of the annual efflux occurs after management events at a relatively short time scale, usage of a GC-based system will be prone to large uncertainties. Paragraph on Page 10, Lines 15-21 will be adjusted.

Changes to the manuscript:
Page 10, Lines 15-21 adjusted to: 'In summary, our comparison of GC vs. QCL fluxes revealed that despite much higher precision, robustness, and temporal resolution in QCL measurements, GC is still a useful method to determine the average campaign $N_2O$ soil efflux. Although single flux values particularly under low exchange regimes did not match well, campaign means and medians were similar to those obtained by the QCL method. Under high exchange regimes, however, flux patterns matched considerably better, but resulted in larger absolute errors when comparing the campaign average, thereby leading to systematic errors (in our case an underestimation) when using the GC method at high $N_2O$ fluxes for the assessment of N balances. However, given the fact that the bulk of the annual efflux occurs after management events at a relatively short time scale (Flechard et al., 2007; Skiba et al., 2013), usage of a GC-based system will be prone to large uncertainties (cf. Fig.7).'

*[R#2.40] FIG 2 add A) and B) to panels and refer to them in the legend*

[AC#2.40] Labels were added and were referred to in the text.

Changes to the manuscript:
Modified Figure 2 with modified caption:

[Figure]

[Figure]

Figure 2. Examples of time series of N$_2$O chamber concentrations during the Braunschweig (Panel A) and Risø campaign (Panel B). Chambers were periodically closed for 60 minutes. Vials were filled with sample air at $t_0$, $t_{20}$, $t_{40}$, and $t_{60}$. The QCL system was operated at a sampling frequency of 10 Hz; plotted are 1-min means.

FIG 4
*[R#2.41] - Refer to "A, B, C and D" before each legend text parts; "Figure 4. a) Comparison of N2O fluxes… b) Linear regression…"*

[AC#2.41] Modified as suggested. Please also notice that Panel D has been changed for a better visualization of lower fluxes and its regression as a result to comment [R#2.26].

Changes to the manuscript:
Modified Figure 4 caption reads:

Figure 4. Panel A: Comparison of N$_2$O fluxes measured on a harvested willow field during the Risø campaign by the QCL system based on a linear model using only the first three minutes of data after chamber closure (filled blue circles) and an exponential model (open red circles) (see text) using either the full 60 minutes (DOY 105.5 to DOY 108.5) or the full 10 minutes of data (DOY <105.5 and DOY >108.5). Panel B: Linear regression analysis of N$_2$O fluxes from

the exponential vs. the linear model. Panel C: Standard errors of fluxes shown in Panel A. Panel D: Same as Panel B, but only for fluxes <200 $\mu$g N m$^{-2}$ h$^{-1}$ with adapted regression.

*[R#2.42] - The legend text should be shortened. Remove phrases such as "Also shown is…" Figure 4b is showing 60-min fluxes plotted against 3 min fluxes.*

[AC#2.42] Modified as suggested.

Changes to the manuscript:
See [AC#2.41].

*[R#2.43] - Please remove the text "Riso campaign 2013, Willow…" from the top of each separate panel and add that part of information into the legend text which is not already there.*

[AC#2.43] Modified as suggested.

Changes to the manuscript:
See [AC#2.41].

*[R#2.44] - Panel B: indicate what are the two lines in the figure? Why are they not direct lines, but show some tiny variation?*

[AC#2.44] The dashed black line is the 1:1 line and the blue solid line is the linear regression fit line. Line labels have been added to Panel B and D. They probably didn't appear as straight lines in the former version, because of the graphical resolution of the figure. This has now been improved.

Changes to the manuscript:
Modified Figure 4 (for caption see [AC#2.41]):

[Figure]

*[R#2.45] - In Fig. 4 and Fig 5, what is the reason to compare 3-min linear and 60-min exponential fluxes? Why not to compare separately the lin vs exp AND 3-min vs 60 min closure times?*

[AC#2.45] In Figures 4 and 5 we explicitly deal with high-resolution measurements of the QCL, which gives us the opportunity to use robust and precise data to compare the application of a linear model simulating short closure time (here 3 minutes) with a non-linear model representing long closure time (here 60 minutes). One of the main aims of the paper is to investigate whether it is suitable to reduce chamber closure time and to apply simple linear regression to calculate the $N_2O$ flux. Through Figure 4 (particularly Panels A and D) and Figure R1 (see above), we demonstrate that the bulk of the flux differences between linear and non-linear models is in an acceptable range, keeping in mind that shorter closure times also have the advantage that plants and soil in the measurement plots are less affected in the long term. Hence, comparing fluxes from a linear model using 3 minutes of data with fluxes from a non-linear model using 60 minutes of data clearly supports our specific aim of the study, while for example applying a linear model to a 60-min data set that reveals obvious curvature (see $\kappa$ values in Figure 3) or applying a non-linear model to only 3 minutes of data that are quasi linear would not give any further insights when investigating whether reducing chamber closure is acceptable or not.

Changes to the manuscript:
None.

FIG 7
*[R#2.46] - Please use A-D notations, not left/right/upper/lower explanations*

[AC#2.46] We added Labels A to D and included them in the figure caption to keep it consistent with Figure 4.

Changes to the manuscript:
Modified Figure 7 with modified caption:

[Figure]

Figure 7. Panels A and B: GC vs. QCL-based $N_2O$ fluxes. Panels C and D: Relationships between standard errors (SE) of $N_2O$ fluxes and the respective flux values. Blue markers indicate QCL data, which are all based on the 3-min linear calculation method. Red markers indicate GC data, which are based on the full 60-min data set. Crosses are plotted for GC data when all criteria for flux calculation using the exponential HMR model were met (see text for details), otherwise circles are plotted indicating the usage of a linear model for flux calculation.

*[R#2.47] - Upper panel: define the lines*

[AC#2.47] Line labels added.

Changes to the manuscript:
See [AC#2.46].

FIG 8
*[R#2.48] - An interesting Figure. What does the error bar denote? Is the diurnal variation significant? Why is the hourly data not shown? Are the points averages from many hours? What was actually the frequency of measurements in both campaigns, I did not find it, but I assumed you measured hourly?*

[AC#2.48] The error bar indicates the standard error of the mean from all flux values in each bin. Each bin contains fluxes from 3-hour periods, i.e. from 00:00 to 03:00, 03:00 to 06:00, 06:00 to 09:00 and so on. The mean values in Figure 8 are plotted in the center of each bin. Fluxes were binned due to irregular starting times of new chamber cycles. In general, a new chamber cycle could be started each full hour, but to get a more robust diurnal pattern, we decided to bin data in the above-mentioned 3-hour containers. While the diurnal variation of $N_2O$ fluxes from the Risø campaign is significant ($p$-value = 0.0059), the diurnal variation found during the Braunschweig campaign is not as the difference between mean minimum and maximum values is lower than the upper flux detection limit of ~2.6 $\mu$g N m$^{-2}$ s$^{-1}$ (*cf.* response to SC3).

Changes to the manuscript:
Information on data handling and significance of the diurnal variation is added to Section 3.4.

References cited in this document:

Bates, D.M. and Watts, D.G.: Nonlinear Regression Analysis and Its Applications, Wiley, 1988.
Drösler, M., 2005. Trace gas exchange and climatic relevance of bog ecosystems, Southern Germany. Doctoral thesis, TU München, 1-182 pp.
Flechard, C.R., Ambus, P., Skiba, U., Rees, R.M., Hensen, A., van Amstel, A., van den Pol-van Dasselaar, A., Soussana, J.-F., Jones, M., Clifton-Brown, J., Raschi, A., Horvath, L., Neftel, A., Jocher, M., Ammann, C., Leifeld, J., Fuhrer, J., Calanca, P., Thalman, E., Pilegaard, K., Di Marco, C., Campbell, C., Nemitz, E., Hargreaves, K.J., Levy, P.E., Ball, B.C., Jones, S.K., van de Bulk, W.C.M., Groot, T., Blom, M., Domingues, R., Kasper, G., Allard, V., Ceschia, E., Cellier, P., Laville, P., Henault, C., Bizouard, F., Abdalla, M., Williams, M., Baronti, S., Berretti, F., Grosz, B.: Effects of climate and management intensity on nitrous oxide emissions in grassland systems across Europe, Agric. Ecosyst. Environ., 121, 135−152, 2007.
Lai, D.Y.F., Roulet, N.T., Humphreys, E.R., Moore, T.R., and Dalva, M.: The effect of atmospheric turbulence and chamber deployment period on autochamber $CO_2$ and $CH_4$ flux measurements in an ombrotrophic peatland, Biogeosciences, 9, 3305−3322, doi:10.5194/bg-9-3305-2012, 2012
Moré, J.J.: The Levenberg-Marquardt algorithm: implementation and theory, in Lecture Notes in Mathematics 630: Numerical Analysis, G.A. Watson (Ed.), Springer: Berlin, 1978, pp.105-116.
Parkin, T.B., Venterea, R. T., and S. K. Hargreaves: Calculating the detection limits of chamber-based soil greenhouse gas flux measurements, J. Environ. Qual., 41, 705−715, doi:10.2134/jeq2011.0394, 2012.

[revised manuscript text omitted]

---

## Referee Report (RR1)

The manuscript has clearly been improved in some parts. Particularly the separation of results and discussion helped a lot. After reading the new version of the manuscript with a much extended discussion, the problematic points in the manuscript show up even more clearly – this is because now one can more easily separate the single experiments and the logics behind them. I have three major points:

1. Which regression method and closure time should be used to get the most accurate estimate of the real flux?

In my opinion, the most apparent pitfall is still the fact that the tests and experiments shown in the paper and the conclusions drawn from them are not well justified, but seem to be set up rather arbitrarily. For example, the question of linear vs non-linear regression is tackled by comparing fluxes calculated with both methods. Fine, but unfortunately you are not able to disentangle what is the effect of the used regression method only, and what part of the difference comes from the different closure times (Fig. 4A). Another example: I do not see where does the recommendation in abstract and conclusion of using chamber closure times of maximum 10 minutes come from? Why not 12 minutes? Or 20 minutes? Where is the basis for that conclusion? I do not find it in results or discussion. Is that coming from the "higher curvature during long chamber closure"? (see point 2 below)

I repeat myself by asking why not to study the effect of regression method separately, by using the same data and the same length of data? And why not to study the effect of closure time separately?  Or why not to study the impact of both by gradually increasing the length of the data, separately for the linear and exponential regression? You do not have plenty of data, which is one of the weaknesses of the paper, but there should be enough to examine the questions of the best regression method and closure time properly.

One more example: the comparison of GC and QCL (Figs. 6 and 7). It is clear that the GC method needs the full 60 minutes, as all the four samples are needed. But what is your argument for selecting exactly the 3-minute length for the QCL data and comparing these two? In other words, can you show that 3 minutes is the best option? And why do you consider the linear fitting most optimal for the 3-minute data?

Same is valid for the point that you remove first 2 minutes. Can you show how much you affect your flux estimate by doing that? I know that in some papers the authors just shortly report that they removed a certain amount of data and used a certain regression method, without justifying anything (e.g. Savage et al. 2014), but as you are already comparing different closure times and regression methods in your paper, why not to do it in a proper way?

So to conclude this point, how do you justify that you use exactly the pre-defined closure times of 3, 10 and 60 minutes? This could be somehow explained by stating that these were the times which you have used and are going to use, and here you just want to show how the fluxes calculated with these closure times relate to each other. But this does not remove the problem that when comparing 3-minute lin fluxes to 60 (10) min exp fluxes you are not able to say whether the difference is due to the closure time or regression method. All you are now showing is that 3-min lin flux produces about the same values as 60-min exp flux, but it is not clear if any of these is the even close to the best estimate of the real flux.

If you, however, want to stick on the figures and results you are currently presenting, you have to give arguments for the relevance of examining such things, for example: "we decided to use 3-min linear calculation (because..?), and as for the GC 60-minute closures are typically used, we wanted to show that 3-min linear is as good as the 60-min exp". But you cannot say based on your analysis shown here that these are the best estimates of the fluxes, or that in other chamber systems it would be a good choice to use 10-minute closures

(as you now do in conclusions). Also, your 3 vs 60 min analysis is not very convincing: you show four flux values >200 µg (Fig. 4B and discussion on p. 12, lines 8-15), three of them destroy your nice correlation, and two out of these three are labelled with different color to indicate that these measurements were showing "steady linear start, followed by a sudden relatively sharp bend..". This emphasizes 1) the small number of flux measurements on which your analysis is based, and 2) there is something strange in your concentration development with high fluxes, if such distortions occur in all high flux values. A question arises, why you want to use such measurements and make generalizations from them, and, last, what is the reason behind them?

**2. Curvature question**

What comes to the questions of curvature and kappa, the conclusion that the closure time had an impact on curvature is definitely not justified by your data (Fig. 3B) – if I have understood right the type of data used for that experiment. You used different measurements (if I understand right the legend text of Fig. 3) for the two box-plots in 3B. Those with 10-min closure were done in different days ("before DOY 105.5 and after DOY 108.5") with highly different flux levels than those with 60-min closure. I do not see, how you can conclude that the differences in kappa were due to closure time, and not for example different flux levels or soil moisture (possibly affecting the curvature) or some other reason. The kappa question should definitely be addressed by using the 60 minute QCL measurements only, and by calculating the kappa for different closure times. E.g., increasing the closure time a few minutes at a time. If you then see that kappa increases (or actually decreases), then you can argue that using longer closure times you get more distorted concentration curves.

However, the question will still remain: what is the reason behind the changing kappa when data gets longer? It might be that it is an inherent tendency of the exp function to result in a steeper increase (or decrease) in $c'(t)$ and $c''(t)$ when t=0, with a smaller amount of data used, because of the higher relative noise in the data due to the shorter closure time. It does not mean, however, that using a shorter closure time is automatically the best choice, but vice versa it could mean that the exponential function causes more uncertainty when using too short data. Can you show which closure time is too short, in terms of kappa?

Abstract line 18: Why it was expected that the curvature was higher with long chamber closure than with 10 minute closure? In chapter 4.1 you state that "This was expected as the rate of transport …declines …because …decline in the vertical concentration gradient". Of course it is expected that the concentration change is non-linear! But this sentence does not justify, why kappa would change with longer closure. Please explain, why it is expected.

In general, the wordings and terms describing curvature in chapter 4.1 are used in a bit sloppy way. Please define clearly what you mean by curvature: that the rate of change in concentration is changing, i.e. $c'(t)$ is changing (which is self-evident), or that the second derivative at t=0 is changing when you use longer closure time.

**3. Conclusions chapter**

Your conclusions section is a summary now, not really concluding anything. There are only two conclusions on lines 6-8 now, related to closure time (unjustified!) and the frequency of raw data (nicely justified). Please rewrite. Also, it is still not clear for me, which fitting method you prefer based on the results of this study? Linear or exponential? Could you indicate this in abstract and conclusions?

**I recommend that the authors carefully consider the points presented in (1) above, and if they decide not to change much, at least they should add a justification and clarification what are the limitations in their analysis. The points (2) and (3) should be taken into account and corrected in the revised version.**

Some detailed comments:

p. 3 lines 31-32: remove this sentence, not needed. A colon within a list followed by a colon is strange, remove.

p. 4 line 1 "density", should it rather be sealing or tightness?

p. 10 line 17 onwards: what is "increase characteristics"? Of what?

p. 13 lines 16-18: This sentence is difficult to understand. Perhaps a word is missing?

Figures 3, 4, 6 and 7: It is still challenging to read and understand the figures and the differences between them, since different closure times and regression methods have been used. I hope this could be solved by plotting different figures as suggested above.

In Fig. 4B: indicate what are the red circles here. Now the reader assumes they are "60 min and 10 min closure", although it has been told in discussion that these are measurements with astrange shape in concentration increase. Are these same data as in 4A? If yes, add 10-min to y-axis legend. If not, why not? Same to 4D, which seems to consist mostly of 10-minute fluxes.  (?)

It seems to me that figures 4B and 7 B are showing contrasting results: in 4B the 3-min linear flux is only half of the 60-min exp flux, while in 7B the 3-min lin flux is twice that of 60-min exp (mostly, since HMR was used). What is this telling us? Impossible to say, because so many variables are changing from figure to figure. This pair of figures crystallizes the confusion the reader have when spending time with the manuscript!!

---

## Author Response (AR2)

**Response to Report #2 from Referee #2 on 'Gas chromatography vs. quantum cascade laser-based N$_2$O flux measurements using a novel chamber design'**

*The manuscript has clearly been improved in some parts. Particularly the separation of results and discussion helped a lot. After reading the new version of the manuscript with a much extended discussion, the problematic points in the manuscript show up even more clearly – this is because now one can more easily separate the single experiments and the logics behind them. I have three major points:*

$\rightarrow$ We thank Referee #2 for taking the time to go through the revised version. However, we have the impression that his/her expectations towards the focus and extent of analyses in our study are somewhat not matching with the aims of the paper, which are clearly outlined on Page 3, Lines 30 ff. While we fully agree with those points from the last round and we again appreciate the very helpful input, we feel that most of the points raised below are not within the intended scope of the present study. Including these points, which surprisingly are now brought up in the second round of revisions, would lead to an entirely new publication. We do not intend to give away the analyses that have already been carried out and think it makes much more sense to keep the story as it has already been presented. See below for details.

*1. Which regression method and closure time should be used to get the most accurate estimate of the real flux?*

*In my opinion, the most apparent pitfall is still the fact that the tests and experiments shown in the paper and the conclusions drawn from them are not well justified, but seem to be set up rather arbitrarily. For example, the question of linear vs non-linear regression is tackled by comparing fluxes calculated with both methods. Fine, but unfortunately you are not able to disentangle what is the effect of the used regression method only, and what part of the difference comes from the different closure times (Fig. 4A).*

$\rightarrow$ It is not the aim of the paper to disentangle the effect of different regression types or closure times. This should be clear by reading the revised version. The aim is to simply COMPARE fluxes with commonly used calculation methods and closure times when using our novel chamber system. The Reviewer is asking for analyses that were never in the scope of this paper.

*Another example: I do not see where does the recommendation in abstract and conclusion of using chamber closure times of maximum 10 minutes come from? Why not 12 minutes? Or 20 minutes? Where is the basis for that conclusion? I do not find it in results or discussion. Is that coming from the "higher curvature during long chamber closure"? (see point 2 below)*

$\rightarrow$ The basis for that recommendation is that (a) the comparison of 3-min (lin) vs. 60-min (exp) flux calculation led to an excellent agreement (Fig.4D), hence short closure time and applying linear flux calculation is applicable without causing large errors, (b) the first 2 minutes after closure should be discarded due to the problems given in the Supplementary Section (mainly pressure fluctuations), which have been widely discussed in Section 4.2, and (c) to have some extra minutes of measurements to be on the safe side when setting the time frame on the data set for flux calculation. Of course for example 8 minutes may be sufficient, but we suggest 10 minutes as a fair compromise between keeping chamber

closure time to a minimum and having a long enough time series of data for appropriate flux calculation. We agree that so far only the minimum of 5 minutes had been mentioned in the text, thus we add 'and a maximum of 10 minutes' on Page 12, Line 23 to name the same number as in the Abstract and Conclusion. We like to point out again that this recommendation is only valid for this particular chamber system under the given conditions. Of course these numbers need to be double-checked for every other campaign and system (which is not too difficult and should be self-evident for everyone who is operating chambers). It is simply not possible to come up with a conclusion that is valid for the whole chamber community as the systems likely behave in a different way every time they are set up in a different location/ecosystems.

*I repeat myself by asking why not to study the effect of regression method separately, by using the same data and the same length of data? And why not to study the effect of closure time separately? Or why not to study the impact of both by gradually increasing the length of the data, separately for the linear and exponential regression? You do not have plenty of data, which is one of the weaknesses of the paper, but there should be enough to examine the questions of the best regression method and closure time properly.*

→ Again, the aims of the study are clearly outlined at the end of the Introduction on Page 3, L.30 ff. All we propose is to 'present a novel chamber design' in order to 'characterize the shape of the concentration increase/decrease', 'compare $N_2O$ fluxes and their associated errors from linear and non-linear regression models', 'test the chamber system under high and low flux conditions and comparing GC vs. QCL-based fluxes', and to 'investigate specific flux characteristics like $N_2O$ uptake and diurnal variation'. That is all we want to show and we think we are doing this in a scientifically valid way. A sensitivity analysis of gradually increasing the length of data for the respective flux calculation has never been intended and is – although surely being interesting – without the scope of this study.

We disagree that the relatively short campaign durations are a weakness of the paper. As mentioned in our last response, the range of test conditions was increased by moving the system to different places where we could investigate very low and high fluxes, high external wind speeds (Risø), moderate wind speeds (Braunschweig), lower and higher temperatures. We think that performing longer tests would not have considerably increased the information with respect to the objectives of the study.

*One more example: the comparison of GC and QCL (Figs. 6 and 7). It is clear that the GC method needs the full 60 minutes, as all the four samples are needed. But what is your argument for selecting exactly the 3-minute length for the QCL data and comparing these two? In other words, can you show that 3 minutes is the best option? And why do you consider the linear fitting most optimal for the 3-minute data?*

→ Of course the three minutes were arbitrarily chosen at the beginning. But again, it is all about finding the best trade-off between having enough data for a stable flux calculation and keeping the chamber closure as short as possible, which is mentioned multiple times in the manuscript (Abstract, Section 4.2, Conclusions). Further, we demonstrate that applying linear regression based on three minutes of data leads to almost the 'same' results ($R^2$=0.93; slope=0.989; Fig.4D) as using 60 minutes and calculating the flux with non-linear regression, which is a strong argument for using linear regression instead of a much more complicated non-linear model where starting parameters need to be initialized and chambers had to be closed for a lengthy time period that negatively impacts the natural conditions of the

measurement spot even more. The recommendation is then to close the chamber for at least 5, but not more than 10 minutes. The reasoning is given in the text (Section 4.2) and in the comment above.

*Same is valid for the point that you remove first 2 minutes. Can you show how much you affect your flux estimate by doing that? I know that in some papers the authors just shortly report that they removed a certain amount of data and used a certain regression method, without justifying anything (e.g. Savage et al. 2014), but as you are already comparing different closure times and regression methods in your paper, why not to do it in a proper way?*

→ We do not see any value in providing an error estimate for 5 % of the occurrences that were discarded anyway following the recommendation of removing the first two minutes of data after chamber closure. What would be the point? The numbers given here do represent this specific chamber system under the given conditions. Every other chamber operator needs to check his own data for those increase/decrease characteristics right after chamber closure. Instead, this study is one of the few that lays open the likely pressure fluctuations by showing the respective figure in the Supplementary.

*So to conclude this point, how do you justify that you use exactly the pre-defined closure times of 3, 10 and 60 minutes? This could be somehow explained by stating that these were the times which you have used and are going to use, and here you just want to show how the fluxes calculated with these closure times relate to each other. But this does not remove the problem that when comparing 3-minute lin fluxes to 60 (10) min exp fluxes you are not able to say whether the difference is due to the closure time or regression method. All you are now showing is that 3-min lin flux produces about the same values as 60-min exp flux, but it is not clear if any of these is the even close to the best estimate of the real flux.*

→ See comments above.

*If you, however, want to stick on the figures and results you are currently presenting, you have to give arguments for the relevance of examining such things, for example: "we decided to use 3-min linear calculation (because..?), and as for the GC 60-minute closures are typically used, we wanted to show that 3-min linear is as good as the 60-min exp". But you cannot say based on your analysis shown here that these are the best estimates of the fluxes, or that in other chamber systems it would be a good choice to use 10-minute closures (as you now do in conclusions).*

→ Yes, for the above-mentioned points, we like to stick with the figures that are already shown. Again, we like to refer to the clearly stated aims of the paper. At no point in our study we claim to present best estimates, nor do we derive any conclusions that are valid or should be applied for other chamber systems. Those points are just simply not the content of this study and we do not understand why Referee #2 insists on giving statements that cannot be formulated with the type of analyses we have conducted.

*Also, your 3 vs 60 min analysis is not very convincing: you show four flux values >200 µg (Fig. 4B and discussion on p. 12, lines 8-15), three of them destroy your nice correlation, and two out of these three are labelled with different color to indicate that these measurements*

*were showing "steady linear start, followed by a sudden relatively sharp bend..". This emphasizes 1) the small number of flux measurements on which your analysis is based, and 2) there is something strange in your concentration development with high fluxes, if such distortions occur in all high flux values. A question arises, why you want to use such measurements and make generalizations from them, and, last, what is the reason behind them?*

→ Why is the 3 vs. 60 min analysis not convincing? Is the Referee expecting an almost perfect fit between the two methods? Instead of discarding the problematic high fluxes, we discuss possible reasons for the mismatch at that range of fluxes (see Section 4.2, P.11, L.30 ff, P.12, L.1-8). We do not agree that such distortion occurs with ALL high flux rates. What is a high flux rate in this context? $N_2O$ fluxes around 50 µg N $m^{-2}$ $h^{-1}$ can also be regarded as 'high' and we show that up to 200 µg N $m^{-2}$ $h^{-1}$ the two methods match pretty well (Fig.4). What is the Referee's point when 4 out of 109 fluxes are 'problematic'? These data are from field campaigns with sensitive devices that are always somewhat error-prone.

*2. Curvature question*

*What comes to the questions of curvature and kappa, the conclusion that the closure time had an impact on curvature is definitely not justified by your data (Fig. 3B) – if I have understood right the type of data used for that experiment. You used different measurements (if I understand right the legend text of Fig. 3) for the two box-plots in 3B. Those with 10-min closure were done in different days ("before DOY 105.5 and after DOY 108.5") with highly different flux levels than those with 60-min closure. I do not see, how you can conclude that the differences in kappa were due to closure time, and not for example different flux levels or soil moisture (possibly affecting the curvature) or some other reason. The kappa question should definitely be addressed by using the 60 minute QCL measurements only, and by calculating the kappa for different closure times. E.g., increasing the closure time a few minutes at a time. If you then see that kappa increases (or actually decreases), then you can argue that using longer closure times you get more distorted concentration curves.*

→ Again, the intention is not a sensitivity study. All we do in Fig.3B is comparing kappa from the long and short closure time. It is true that fluxes during the 10-min closure were considerably higher than those during the long closure (the different periods were chosen with respect to the availability of vials for the GC system). The results were somewhat expected. From theory one would expect higher kappa for higher fluxes (since it is the derivative of the flux) and the findings in Fig.3A show this experimentally as there is a tendency of high kappa mainly occurring at high fluxes (which could not be the case when flux magnitudes would be driving kappa in Fig.3B as the Referee assumes). But the intriguing thing is exactly the fact that although fluxes were higher under shorter closure, absolute kappa values were lower than under low fluxes and longer closure. Hence, the higher kappa values observed in Fig.3B for longer closure time were not due to higher flux magnitudes.

*However, the question will still remain: what is the reason behind the changing kappa when data gets longer? It might be that it is an inherent tendency of the exp function to result in a steeper increase (or decrease) in c'(t) and c''(t) when t=0, with a smaller amount of data used, because of the higher relative noise in the data due to the shorter closure time. It does*

*not mean, however, that using a shorter closure time is automatically the best choice, but vice versa it could mean that the exponential function causes more uncertainty when using too short data. Can you show which closure time is too short, in terms of kappa?*

→ This is a very interesting question. The signal to noise ratio is definitely lower for shorter time series. Also the point of saturation is less visible in shorter time series leading to a higher uncertainty in the decay parameter $k$ and $c_{max}$, and hence also kappa, when fitting the exponential curve. This would surely be worth looking at in a sensitivity study by using the 60 min dataset and then refitting the curve for shorter and shorter time periods, but this is not within the scope of this paper. Also, we are surprised that the Referee is talking about 'best choice' methods, which have never been claimed to be tested for in this paper. All we investigated is that shorter closure and applying linear regression is just as good as longer closure with non-linear regression with the former having the advantage of keeping the soil spot less disturbed.

*Abstract line 18: Why it was expected that the curvature was higher with long chamber closure than with 10 minute closure? In chapter 4.1 you state that "This was expected as the rate of transport …declines …because …decline in the vertical concentration gradient". Of course it is expected that the concentration change is non-linear! But this sentence does not justify, why kappa would change with longer closure. Please explain, why it is expected.*

→ In our study, we observed higher kappa values for longer closure times. Please see answer to the previous question for potential reasons. We changed the first paragraph of section 4.1 including the sentences quoted by the referee to be more precise about the characteristics of kappa (see next point).

*In general, the wordings and terms describing curvature in chapter 4.1 are used in a bit sloppy way. Please define clearly what you mean by curvature: that the rate of change in concentration is changing, i.e. c'(t) is changing (which is self-evident), or that the second derivative at t=0 is changing when you use longer closure time.*

We agree with the referee that kappa should be introduced more explicitly and changed the first paragraph of section 4.1. as follows:

P.10, L.17: 'The high time resolution of QCL data allowed for a closer look at the shape of the concentration increase. The general form of the curve is determined by the rate of transport of a diffusing trace gas into the chamber headspace, which declines throughout deployment because any increase in the headspace concentration results in a corresponding decline in the vertical concentration gradient driving that transport (Rolston, 1986; Hutchinson et al., 2000; Livingston et al., 2006). The change in the rate of transport is the initial curvature kappa, i.e. the second derivative of the concentration change at $t$=0.'

*3. Conclusions chapter*

*Your conclusions section is a summary now, not really concluding anything. There are only two conclusions on lines 6-8 now, related to closure time (unjustified!) and the frequency of raw data (nicely justified). Please rewrite. Also, it is still not clear for me, which fitting method you prefer based on the results of this study? Linear or exponential? Could you indicate this in abstract and conclusions?*

→ Correct, the Section briefly summarizes the results and CONCLUDES some take home messages in the form of recommendations. This is commonly done in that way when finishing a scientific paper. We do not think this is too much of a summary, although one could rename it into 'Concluding remarks'. Regarding closure time and its justification, see points above. We thank the Referee for pointing out that a recommendation for the calculation method is missing. This has been added to both Abstract and Conclusion.

*I recommend that the authors carefully consider the points presented in (1) above, and if they decide not to change much, at least they should add a justification and clarification what are the limitations in their analysis. The points (2) and (3) should be taken into account and corrected in the revised version.*

*Some detailed comments:*

*p. 3 lines 31-32: remove this sentence, not needed. A colon within a list followed by a colon is strange, remove.*
→ Done.

*p. 4 line 1 "density", should it rather be sealing or tightness?*
→ Modified to 'sealing'.

*p. 10 line 17 onwards: what is "increase characteristics"? Of what?*
→ Modified to '…allowed for a closer look at the shape of the concentration increase…'

*p. 13 lines 16-18: This sentence is difficult to understand. Perhaps a word is missing?*
→ Thanks for the catch. Sentence modified to: 'The fact that higher fluxes in our study were associated with lower standard errors and accepted HMR application is corresponding well with $\kappa$ findings in Section 3.1 indicating that higher curvature in $c$(t) coincided with higher fluxes (Figure 3B).

*Figures 3, 4, 6 and 7: It is still challenging to read and understand the figures and the differences between them, since different closure times and regression methods have been used. I hope this could be solved by plotting different figures as suggested above.*
→ We think that the analyses and methods in this paper are clearly described and the plotted data are well-arranged in Figures 3 to 7.

*In Fig. 4B: indicate what are the red circles here. Now the reader assumes they are "60 min and 10 min closure", although it has been told in discussion that these are measurements with astrange shape in concentration increase. Are these same data as in 4A? If yes, add 10-min to y-axis legend. If not, why not? Same to 4D, which seems to consist mostly of 10-minute fluxes. (?)*

→ Description for red circles in Panel B added. Yes, 4A and 4B present the same data, thus we added '10-min' to the y-axis label. No, 4D does not mostly consist of 10-minute fluxes, because only fluxes up to 200 µg N m$^{-2}$ h$^{-1}$ are shown, hence the lower ones are included that have been measured on DOY<105.5 and DOY>108.5 where closure time was 60 minutes (see Discussion and respective points above).

*It seems to me that figures 4B and 7 B are showing contrasting results: in 4B the 3-min linear flux is only half of the 60-min exp flux, while in 7B the 3-min lin flux is twice that of 60-min exp (mostly, since HMR was used). What is this telling us? Impossible to say, because so many variables are changing from figure to figure. This pair of figures crystallizes the confusion the reader have when spending time with the manuscript!!*

→ We really do not understand the confusion. Figure 4B shows data from QCL only. Figure 7B compares QCL with GC (!) data. This is clearly described in the Figure captions.

[revised manuscript text omitted]